# An allosteric modulator activates BK channels by perturbing coupling between Ca²⁺ binding and pore opening

Guohui Zhang [1,9], Xianjin Xu[2,3,4,5,9], Zhiguang Jia[6,7,9], Yanyan Geng[8], Hongwu Liang[1], Jingyi Shi[1], Martina Marras [1], Carlota Abella[1], Karl L. Magleby[8], Jonathan R. Silva [1]✉, Jianhan Chen [6,7]✉, Xiaoqin Zou [2,3,4,5]✉ & Jianmin Cui [1]✉

BK type Ca²⁺-activated K⁺ channels activate in response to both voltage and Ca²⁺. The membrane-spanning voltage sensor domain (VSD) activation and Ca²⁺ binding to the cytosolic tail domain (CTD) open the pore across the membrane, but the mechanisms that couple VSD activation and Ca²⁺ binding to pore opening are not clear. Here we show that a compound, BC5, identified from in silico screening, interacts with the CTD-VSD interface and specifically modulates the Ca²⁺ dependent activation mechanism. BC5 activates the channel in the absence of Ca²⁺ binding but Ca²⁺ binding inhibits BC5 effects. Thus, BC5 perturbs a pathway that couples Ca²⁺ binding to pore opening to allosterically affect both, which is further supported by atomistic simulations and mutagenesis. The results suggest that the CTD-VSD interaction makes a major contribution to the mechanism of Ca²⁺ dependent activation and is an important site for allosteric agonists to modulate BK channel activation.

BK type Ca²⁺ activated K⁺ channels are important in regulating neural excitability[1–3], neurotransmitter release[4], and muscle contraction[5,6]. Aberrant function of BK channels is associated with various human diseases such as neurological disorders[7–9], hypertension[10] and impaired urinary control[11]. The function of BK channels relies on their activation by both elevated intracellular Ca²⁺ and depolarization of the membrane voltage. The opening of BK channels provides a negative feedback mechanism to hyperpolarize the membrane and reduce intracellular Ca²⁺ by deactivating voltage-dependent Ca²⁺ channels[12]. The Ca²⁺- and voltage-dependent activation of BK channels has been an important model system for understanding allosteric mechanisms of ion channel gating[13]. Unlike canonical voltage gated K⁺ channels, such as Shaker, BK channels exhibit an intrinsic open probability ($P_o \sim 10^{-6}$) in the absence

of stimuli. Ca²⁺ binding or voltage sensor activation allosterically activates the channel by favoring the open state[13,14]. The mechanisms of voltage and Ca²⁺-dependent BK channel activation have been described by an allosteric model[13], which has successfully accounted for the functional properties of BK channel activation, pharmacology[15], modulation by regulatory subunits[16] and mutations[17,18]. The molecular bases of these allosteric activation mechanisms are not clear. Understanding these molecular mechanisms may help reveal the physiological role of BK channels, facilitate the development of therapeutic agents that modulate BK channel activation, and elucidate fundamental mechanisms of allosteric modulation.

BK channels are formed by four Slo1 subunits with structural domains that are associated with distinct functions[19–21] (Fig. 1a). In the

¹Department of Biomedical Engineering, Center for the Investigation of Membrane Excitability Disorders, Cardiac Bioelectricity and Arrhythmia Center, Washington University, St. Louis, MO, USA. ²Dalton Cardiovascular Research Center, University of Missouri – Columbia, Columbia, MO, USA. ³Department of Physics and Astronomy, University of Missouri – Columbia, Columbia, MO, USA. ⁴Department of Biochemistry, University of Missouri – Columbia, Columbia, MO, USA. ⁵Institute for Data Science and Informatics, University of Missouri – Columbia, Columbia, MO, USA. ⁶Department of Chemistry, University of Massachusetts, Amherst, MA, USA. ⁷Department of Biochemistry and Molecular Biology, University of Massachusetts, Amherst, MA, USA. ⁸Department of Physiology and Biophysics, University of Miami Miller School of Medicine, Miami, FL, USA. ⁹These authors contributed equally: Guohui Zhang, Xianjin Xu, Zhiguang Jia. ✉e-mail: jonsilva@wustl.edu; jianhanc@umass.edu; zoux@missouri.edu; jcui@wustl.edu

membrane, a central pore is formed by the transmembrane segments S5–S6 from all four subunits. Surrounding the pore are the voltage sensor domains (VSDs) from the four subunits, comprised of the transmembrane segments S1–S4. Slo1 also contains an additional transmembrane segment S0. The cytosolic tail domain (CTD) of each Slo1 subunit harbors two $Ca^{2+}$-binding sites, and the four CTDs, each containing two structural motifs known as the regulator of potassium

conductance (RCK), form a ring-like structure, known as the gating ring. Studies have shown that $Ca^{2+}$ and voltage activate BK channels with distinct molecular mechanisms such that depolarization can activate the channel in the absence of $Ca^{2+}$ binding[22], while $Ca^{2+}$ binding can activate the channel when the voltage sensor is at the resting state[13,14,23]. In these activation mechanisms $Ca^{2+}$ or voltage alter the conformation of their respective sensors, and the conformational

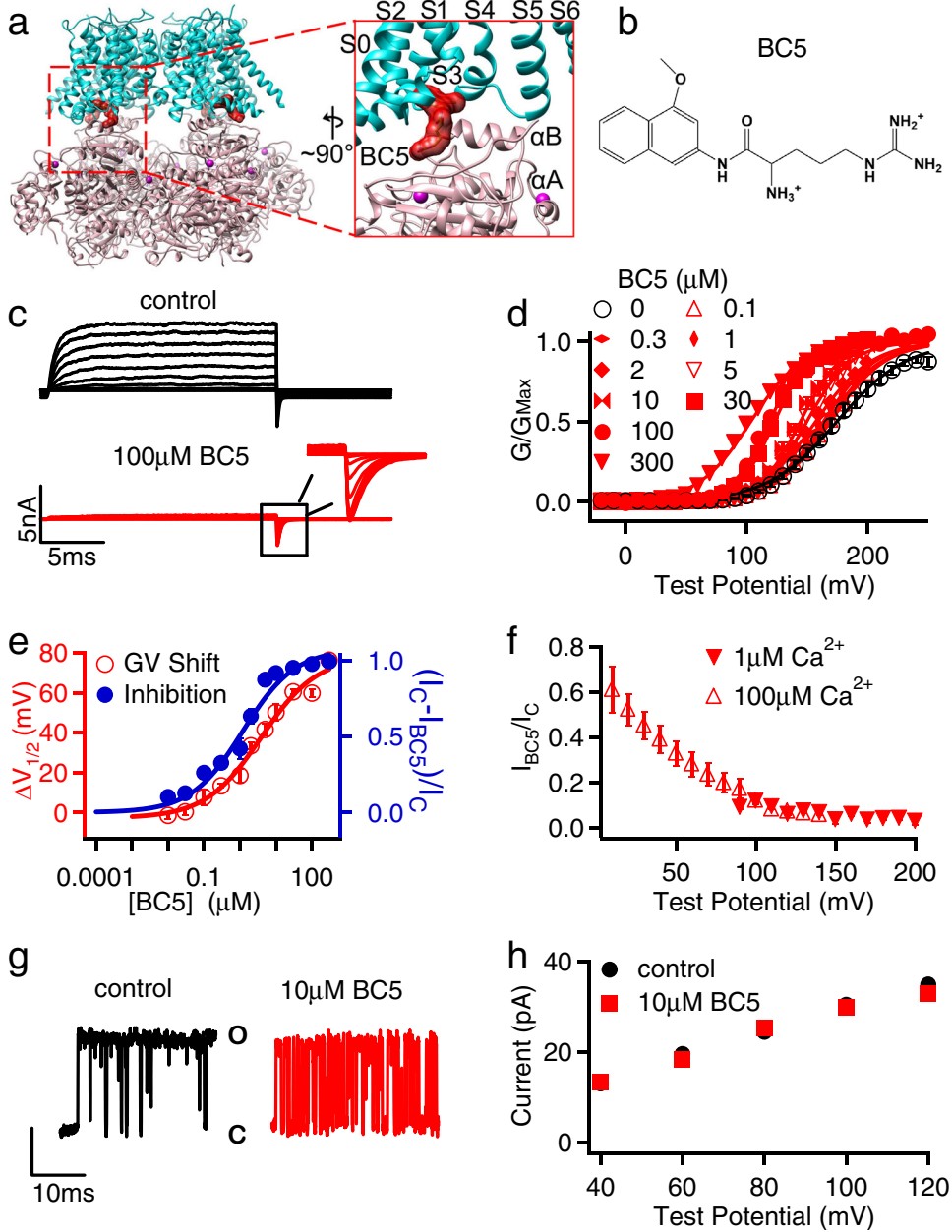

**Fig. 1 | BC5 effects on BK channels. a** *hSlo1* Channel cryo-EM structure (PDB: 6v38) docked with BC5. The membrane spanning region (S0-S6) from one subunit and the N-terminus of the cytosolic tail domain (CTD) from a neighboring subunit are amplified in the inset. The CTD and VSD are colored pink and cyan, respectively. The BC5 is colored in red and represented as molecular surface. **b** BC5 chemical structure. **c** Current traces of BK channels (control, black) and with BC5 (100 μM, red) at various testing voltages from −30 to 250 mV with 20 mV increment. The potential before and after testing pulses was −80 mV. Inset: tail currents after the testing pulses. **d** GV relationship for WT BK channels at various BC5 concentration. Solid lines are fits to the Boltzmann relation. All data were collected from independent patches, *n* = 3–12. For detailed *n* for each data point, please see the "Reproducibility" section in "Methods." **e** The shifts of GV relationships

(red, $V_{1/2}$: voltage where the GV relation is half maximum) and outward macroscopic current inhibition (blue, $I_C$: control current, $I_{BC5}$: current in BC5) depend on BC5 concentration, with EC$_{50}$ of 2.5 ± 0.3 μM and IC$_{50}$ of 1.1 ± 0.1 μM respectively. *n* = 3–12, exact n for each data point is shown in the "Reproducibility" section in "Methods." **f** Inhibition of the outward macroscopic current by 100 μM BC5 versus voltage in the presence of either 1 μM (filled, *n* = 3) or 100 μM (hollow, *n* = 4) free $Ca^{2+}$. **g** Representative single-channel current recordings at 100 mV and in 100 μM free $Ca^{2+}$ as control (left and black) and with 10 μM BC5 (right and red) from an inside out patch in symmetrical 160 mM KCl. O: open; C: closed. **h** Voltage dependence of single-channel current amplitudes (i-V), *n* = 3. The data points represent the mean ± SEM for the whole paper.

change is then coupled to the opening of the activation gate via pathways of intramolecular interactions in the channel protein[24]. Although the sensors for $Ca^{2+}$ and voltage[25] and the properties of the activation gate[25,26] have been identified, the allosteric coupling mechanisms between the sensors and the activation gate in BK channels are still under investigation[27,28]. One of the long-standing issues is to identify the paths of intramolecular interactions within the channel protein by which cytosolic $Ca^{2+}$ binding or voltage sensor activation opens the channel pore across the membrane[19,20,23,27–33]. Another issue is that both $Ca^{2+}$ and voltage open the same activation gate. Therefore, the respective paths for mediating voltage and $Ca^{2+}$-dependent channel opening must merge eventually. Although $Ca^{2+}$ and voltage activate BK channels via distinct mechanisms, the two activation mechanisms influence each other allosterically[13,34]. Where and how the allosteric paths for $Ca^{2+}$- and voltage-dependent activation merge remains a fundamental question.

In BK channels while the CTD serves as the $Ca^{2+}$ sensor it was also shown to contribute to voltage-dependent gating[28,31,34], and CTD conformation was altered by voltage-dependent activation[35]. For $Ca^{2+}$ binding at the CTD and for the VSD activation to open the pore, interactions among the CTD, VSD and the pore domains are essential. Although the structural domain interactions that are required for voltage and $Ca^{2+}$-dependent activation have been intensively studied, mechanistic roles of these domain interactions remain poorly understood. In the Slo1 subunit the CTD is covalently connected to the S6 by a peptide of 15 amino acids, known as the C-linker. A previous study showed that deletion or insertion of C-linker residues altered $Ca^{2+}$- and voltage-dependent activation in a manner that correlated with the C-linker length, suggesting that $Ca^{2+}$ binding to the CTD may open the channel by "tugging" the C-linker[29]. On the other hand, the CTD also has a non-covalent interaction with the VSD of its neighboring subunit (Fig. 1a). This interaction was first revealed by the discovery that the CTD and VSD form a close interface because residues from both the CTD and VSD of the neighboring subunits form a $Mg^{2+}$-binding site, which differs from the $Ca^{2+}$-binding sites in CTD. $Mg^{2+}$ binding activates the channel by enhancing voltage-dependent activation via an electrostatic interaction with the voltage sensor[36]. The interactions between CTD and VSD were then more clearly shown by BK channel structures[19,20]. Based on the structural evidence, it was hypothesized that the CTD-VSD interaction may contribute to the coupling between $Ca^{2+}$ binding and pore opening[30]. This notion was strongly supported by the fact that the comparison between the structure of the BK channel with $Ca^{2+}/Mg^{2+}$ bound and that of metal-free showed a large $Ca^{2+}/Mg^{2+}$-dependent change in the interface between CTD and VSD[19,20]. This hypothesis is also consistent with an earlier finding that the region of the CTD located close to the VSD is important in determining the different $Ca^{2+}$ sensitivities among various BK channel homologs[37]. Furthermore, mutations at the CTD-VSD interface were also found to alter $Ca^{2+}$ sensitivity[27,38]. However, these mutations either had small effects on $Ca^{2+}$-dependent activation[38] or, besides reducing $Ca^{2+}$-dependent activation, also altered voltage-dependent activation and resulted in an increase of the intrinsic open probability of the channel[27]. Did these mutations alter the $Ca^{2+}$-dependent activation or did some of these mutations alter the voltage-dependent activation by affecting the voltage sensor, which then indirectly affect $Ca^{2+}$-dependent activation? Thus, whether the CTD-VSD interaction as revealed by structural data mediates coupling of $Ca^{2+}$ binding to pore opening remains unclear.

In this study we use in silico screening to identify a compound that binds to a site at the CTD–VSD interface and enhances BK channel activation. We find that the compound specifically enhances $Ca^{2+}$-dependent activation by perturbing the pathway for coupling between $Ca^{2+}$ binding and pore opening. These results support the idea that the non-covalent CTD–VSD interaction makes a major contribution to the coupling between $Ca^{2+}$ binding and pore opening that is independent of voltage-dependent activation, and that the CTD–VSD interface is an important site for allosteric agonists to enhance $Ca^{2+}$-dependent activation of BK channels.

## Results

In order to probe if the VSD–CTD interaction of BK channels is required for $Ca^{2+}$-dependent activation, we initiated an in silico search for compounds that may interfere with BK channel $Ca^{2+}$-dependent activation. Specifically, we docked a subset of the Available Chemical Database (ACD, Molecular Design Ltd.) of about $4 \times 10^4$ compounds, in which each compound carries one or two charged groups with a net charge of 0, ±1, or ±2, to a site near the VSD-CTD interface (Fig. 1a, Supplementary Fig. 1). Patch-clamp recordings of BK channels expressed in *Xenopus* oocytes with and without the candidate compounds from the in silico screening showed that one of the compounds, BC5 applied at the cytosolic side of the membrane patch (Fig. 1b), activated the channel as measured from the inward tail currents at a voltage (−80 mV) more negative than the equilibrium potential of the $K^+$ (0 mV) (Fig. 1c, d and Supplementary Fig. 2a). BC5 shifted the voltage-dependent activation (GV relation) of BK channels to more negative voltages (Fig. 1d), with the concentration at the half-maximum GV shift ($EC_{50}$) of $2.5 \pm 0.3 \mu M$ (Fig. 1e). In addition, BC5, applied in the cytosolic side of the membrane patch, also inhibited the outward BK currents at positive voltages (Fig. 1c and Supplementary Fig. 2b).

We found that the activation and inhibition of BK channels by BC5 were independent molecular processes. BC5 inhibited the channel at a lower concentration than activation, with the half-maximum inhibition concentration ($IC_{50}$) of $1.1 \pm 0.1 \mu M$ (Fig. 1e, Supplementary Fig. 2b). The inhibition was dependent on voltage, with weaker inhibition at less positive potentials (Fig. 1f) and no inhibition at a negative potential (Fig. 1c). These characteristics of inhibition are consistent with the mechanism that the positively charged BC5 acts as a pore blocker from the cytosolic side, and can be flushed out by negative voltages and inward $K^+$ currents, reminiscent of $Ca^{2+}$ and $Mg^{2+}$ block of the channel[39,40]. Single channel recordings showed that BC5 caused brief closures during channel opening without altering the single channel conductance (Fig. 1g, h), consistent with the mechanism that BC5 acts as a discrete pore blocker with relatively fast on- and off-rates, but not fast enough to be a fast blocker that reduces single channel conductance. In addition, unlike BC5 activation, BC5 inhibition of the channel showed little dependence on $Ca^{2+}$ (Fig. 1f) or $Mg^{2+}$ (Supplementary Fig. 3a, b), and the mutations that reduced BC5 activation had little effect on BC5 inhibition (Supplementary Fig. 4a, b). These results suggest that BC5 inhibition of the channel is an off-target effect, namely, in addition to interacting with our target docking site for channel activation (Fig. 1a), BC5 may interact with the pore (Supplementary Fig. 4c) to block the outward $K^+$ currents.

Our target docking site of BC5 in the BK channel is located near the $Mg^{2+}$-binding site for channel activation, which is formed by residues from both the VSD and CTD[36,41] (Fig. 2a). As shown in Fig. 2a, d, residue E399 is shared by both the $Mg^{2+}$- and BC5-binding sites. If BC5 interacts with the target docking site to activate the channel, we expected that an electrostatic repulsion between $Mg^{2+}$ and the charges in BC5 and a direct competition between $Mg^{2+}$ and BC5 for binding would reduce BC5 activation. Consistent with this idea, we found that BC5 at the same concentration (100 μM) induced less shift in GV relations in the presence of 10 mM $Mg^{2+}$ (Fig. 2b, c and Supplementary Fig. 3a). Likewise, a charge reversal mutation of the $Mg^{2+}$-binding residue, D99R[19], also reduced BC5 activation of the channel (Fig. 2c), although D99 is not part of the BC5-binding pocket, supporting that BC5 binds in the vicinity of the $Mg^{2+}$-binding site.

To further characterize BC5 interactions with the target docking site, 200 ns molecular dynamics (MD) simulations of the open and

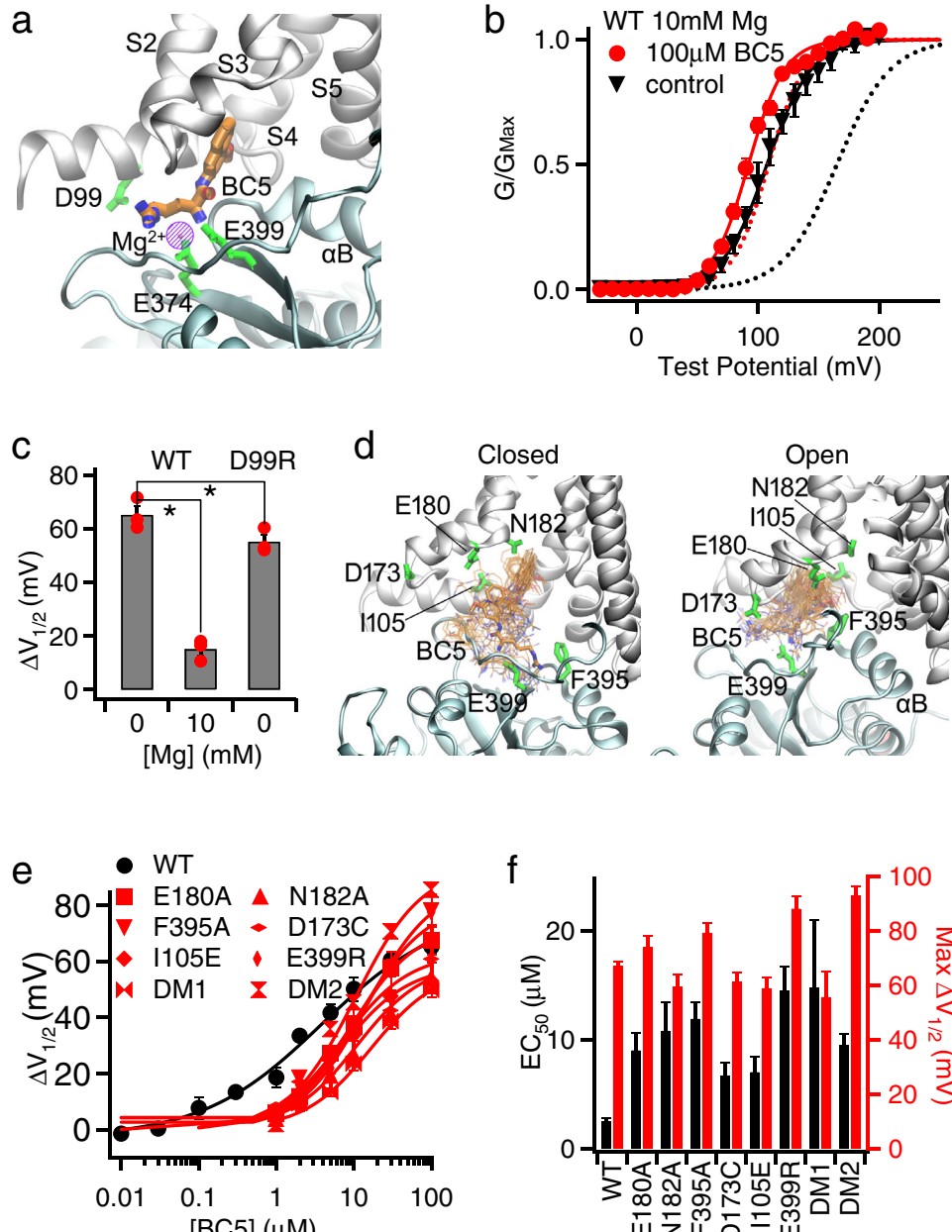

**Fig. 2 | BC5 interactions with the targeted site at the CTD-VSD interface. a** The cryo-EM structure of *hSlo1* (PDB: 6v38) with BC5 docked to its targeted site. A similar region as in Fig. 1a is shown. The VSD and CTD are colored silver and cyan, respectively. $Mg^{2+}$-binding residues (D99, E374, E399) are represented as green sticks. The $Mg^{2+}$ position is highlighted using the purple circle, which is hatched to indicate that $Mg^{2+}$ and BC5 do not bind at the same time. **b** GV relationship of BK channels and Boltzmann fits (solid lines) in 10 mM $Mg^{2+}$. $n = 3$ for control and $n = 4$ for 100 μM BC5. For comparison, the dashed lines represent the GV relationships of the control (black) and 100 μM BC5 (red) in the absence of $Mg^{2+}$, taken from Fig. 1d. **c** GV shifts caused by 100 μM BC5. For WT the GV shifts in 0 ($n = 3$) and 10 mM ($n = 3$) $Mg^{2+}$ are plotted, and for the D99R ($n = 4$) mutant, GV was measured in 0 $Mg^{2+}$. *: p =0.0002 and 0.0475 respectively, unpaired Student's t-test. Data points are shown

in solid circles. **d** Dynamic interactions of BC5 in the binding pocket of metal-free (left) and $Ca^{2+}$-bound (right) BK channels. Snapshots of BC5 (thin sticks) were extracted from the atomistic simulation trajectories every 10 ns and superimposed onto the last frame of *sim 1* and *7* (Supplementary Table 1). As a reference, representative BC5 binding poses (closed: at 180 ns from sim 7; open: at 102 ns from sim 1) are also shown in thick bonds. **e** GV shifts of mutant BK channels in response to BC5. $n = 3$–12, exact n for each data point is shown in the "Reproducibility" section in "Methods." The WT curve is taken from Fig. 1e. **f** Maximal G-V shift ($\triangle V_{1/2}^{max}$) and $EC_{50}$ of BC5 for mutations and the WT. DM1: double mutation E180A/N182A; DM2: I105E/E399R. The data were obtained from fittings of the dose response in Fig. 2e using Hill equation. The standard errors of $EC_{50}$ and $\triangle V_{1/2}^{max}$ were estimated directly from the SEMs of the input data and validated by numerical simulations.

closed states of BK channel with and without docked BC5 were performed in explicit membrane and water (see "Methods") (Supplementary Table 1). The results showed multiple binding modes between BC5 and the channel that interchanged dynamically with time, and in the process BC5 interacted with different residues (Fig. 2d). As summarized in Supplementary Table 2, the hydrophobic naphthalenyl group of BC5 was buried in contact with the intracellular side of S0, S1,

S3, and S4 as well as the residue F395 from αB of CTD. The charged arginine part of BC5 interacted with the negatively charged residue E399 (Fig. 2d). The binding poses of BC5 were more narrowly confined in the open conformation, suggesting that BC5 binding is stronger and may favor the open state of BK channels. We mutated the BC5 interacting residues individually or in combination to alanine or other amino acids that changed the charge of the BC5 interacting residue,

and measured dose responses of the mutant channel activation to BC5 (Fig. 2e). These mutant channels showed increased $EC_{50}$ (Fig. 2f), suggesting that the mutations reduced BC5 binding by interrupting the interactions between those residues and BC5. Since BC5 binds to the channel with multiple modes, a mutation to a single residue in the binding pocket may not be sufficient to abolish BC5 binding. Instead, the mutation may prevent some of the binding modes but reinforce others. This may have happened to some of the mutations, which, while enhancing EC50, also increased the slope of the BC5 dose response and the maximum channel activation (Fig. 2e, f). To further disrupt BC5 binding, we made double, triple or quadruple mutations that combined these individual mutations. However, except for the two shown in Figs. 2e and 2f, the combined mutations abolished channel function. These results suggest that this pocket of residues are important for channel function. Consistently, we found that some of the individual mutations altered channel function (Supplementary Fig. 5). These results support that BC5 enhances BK channel activation by binding to the target docking site.

BK channels are activated by voltage and $Ca^{2+}$ with distinct mechanisms[7]. Thus, we asked whether BC5 shifted the GV relation of BK channels by modulating the voltage or $Ca^{2+}$-dependent activation mechanism. To address this question, we first examined whether BC5 modulated VSD activation in BK channels by measuring gating currents with and without BC5 (Fig. 3a). Unlike the GV relationship, the voltage dependence of gating charge movement (QV) was not shifted to more negative potentials (Fig. 3a). Furthermore, measurements of intrinsic openings of the channel at negative voltages ($<-100$ mV) where the voltage sensors were at the resting state[14,42] showed that BC5 enhanced open probability (Fig. 3b). These results suggest that BC5 did not activate the channel by altering the voltage-dependent mechanism. Since the BC5- and $Mg^{2+}$-binding sites are located closely and share E399 (Fig. 2a), we examined if BC5 activates the channel via a similar mechanism as $Mg^{2+}$ by modulating VSD activation[36]. $Mg^{2+}$ activates BK channels by interacting with the voltage sensor when the VSD is at the activated conformation[43–45], which results in a reduction of the off-gating current amplitude at the end of the depolarizing voltage pulse[46]. The $Mg^{2+}$-VSD interaction also shifts GV relation to more negative potentials. However, $Mg^{2+}$ cannot alter open probabilities at negative potentials when the VSD is not activated[46], differing from the BC5 effect (Fig. 3b). Unlike $Mg^{2+}$, BC5 also had no effect on the amplitude of off-gating currents (Fig. 3a). Therefore, BC5 did not alter voltage-dependent activation indirectly by a mechanism similar to that of $Mg^{2+}$. Previous studies show that $Ca^{2+}$ binding also shifts the GV relation to more negative potentials and enhances open probability at negative potentials[13,23]. Interestingly, the amount of GV shift and the enhancement of open probability at negative voltages caused by 100 μM BC5 were comparable to those caused by 1 μM $Ca^{2+}$[22,23], suggesting that BC5 activated the channel by specifically modulating the $Ca^{2+}$-dependent mechanism.

To further examine the BC5 regulation of the $Ca^{2+}$-dependent activation mechanism, we measured BC5 activation of BK channels at various $Ca^{2+}$ concentrations. We found that $Ca^{2+}$ reduced BC5 activation and the reduction was dependent on $Ca^{2+}$ concentration. At 100 μM $Ca^{2+}$, which saturated $Ca^{2+}$-dependent activation[11], BC5 no longer activated the channel (Fig. 3c, d and Supplementary Fig. 6). The effect of $Ca^{2+}$ on BC5 activation is specifically associated with $Ca^{2+}$-dependent activation of the channel, because mutations that abolished $Ca^{2+}$-binding sites in the CTD for channel activation[47] also abolished $Ca^{2+}$ effects on BC5 activation of the channel (Fig. 3e, Supplementary Fig. 7). We studied a truncated BK channel in which the entire CTD was replaced with a short exogenous peptide, known as the Core-MT[48]. The Core-MT channel was no longer activated by $Ca^{2+}$ because the deletion of all $Ca^{2+}$-binding sites. However, it was still activated by BC5, and 100 μM $Ca^{2+}$ had no effect on BC5 activation

(Fig. 3f and Supplementary Fig. 8a–c). These results suggest that $Ca^{2+}$ competed with BC5 for activating BK channels.

$Ca^{2+}$-binding sites within the CTD are not spatially close to the BC5-binding site at the VSD-CTD interface (Fig. 1a), and further, BC5 activated the mutant channels in which $Ca^{2+}$-binding sites or the entire CTD were eliminated (Fig. 3e, f). Therefore, BC5 likely does not interact with $Ca^{2+}$ binding directly, but rather the competition between $Ca^{2+}$ and BC5 is allosteric. In other words, $Ca^{2+}$ binding to the channel may alter the conformation at the BC5-binding site. For BC5 to open the activation gate independently of voltage on one hand and compete with $Ca^{2+}$ binding on the other hand, it may interact with the molecular pathway that couples $Ca^{2+}$ binding to channel opening. That is, BC5 perturbs the pathway to promote pore opening, which can be overwhelmed by a stronger perturbation at high $Ca^{2+}$ concentrations due to $Ca^{2+}$ binding. To further explore this idea, we performed dynamic network (Fig. 4) and information flow analysis (Supplementary Fig. 9) to examine the coupling pathway from the $Ca^{2+}$-binding residue D367 to the pore lining F315. Specifically, the network was constructed in which each node represented a single residue (or BC5) and the lengths of edges captured the dynamic correlation calculated from the MD trajectories (see "Methods"). Nodes that were more strongly coupled would have shorter connecting edges. We then calculated a set of "optimal" and "suboptimal" paths between the selected residues, which were the shortest paths and thus the most strongly coupled[49]. The top 20 paths with the strongest coupling strengths are shown in Fig. 4a, b (green sticks) to illustrate the allosteric coupling pathways connecting the $Ca^{2+}$-binding site (D367) to the pore residues (F315) in the same subunit and in a neighboring subunit. The $Ca^{2+}$-binding site is in the CTD, while the activation gate is located in the membrane spanning domain. In both open and closed conformations, the allosteric pathway crossed these domains via two major branches, with one branch via the non-covalent VSD-CTD interface from αA/αB in the CTD to the S4-S5 linker/S6 in the VSD, and the other branch via the covalent C-linker to S6. To complement the pathway analysis, we analyzed the information flow[50,51] between D367 ("source") and F315 ("sink"), which quantifies the contributions of all the residues to dynamic coupling. The results show that the information flow through VSD-CTD interface is higher or similar to that through the C-linker/S6 connection (Supplementary Fig. 9), suggesting that this interface is critical for the coupling between $Ca^{2+}$ binding and pore opening in addition to the C-linker/S6 pathway[29]. Importantly, the network analysis shows that BC5 lies adjacent to the key allosteric pathways at the VSD-CTD interface, interacting with residues F395 and E399, which are part of both the BC5-binding site and the allosteric pathway in the closed structure bound with BC5 (Fig. 4a). In addition, in other closed and open structures F395 and E399 are in the vicinity of the allosteric pathway within a distance of < 6 Å (Fig. 4a, b and Supplementary Table 2). These results suggest that BC5 binding likely perturbs the allosteric pathway to interact with both $Ca^{2+}$ binding and pore opening.

We performed a mutational scanning of selected residues on or in the vicinity of the allosteric pathway and found that the mutation T245W reduced whereas E219R enhanced the sensitivity of the channel to $Ca^{2+}$ for activation[18] (Fig. 4c, d). T245 is part of the allosteric pathway, while E219 is in the vicinity of the allosteric pathway in the open state with or without BC5 (Fig. 4b). E219 and T245 are in S4 and S5, respectively, and are downstream from BC5 in the allosteric pathway, being closer to the activation gate (Fig. 4a, b). If BC5 perturbs the allosteric pathway to activate the channel as shown above (Fig. 4a, b; Supplementary Fig. 9) we expected that these two mutations would alter BC5 activation as well. Indeed, similar to the effects on $Ca^{2+}$-dependent activation, T245W reduced BC5 activation of the channel such that BC5 shifted the GV less in the mutant than in the WT channel (Fig. 4c, d), while E219R enhanced BC5 activation (Fig. 4c, d). These results support the notion that the allosteric pathway couples $Ca^{2+}$ binding to pore opening (Fig. 4a, b) and that BC5 activates the channel

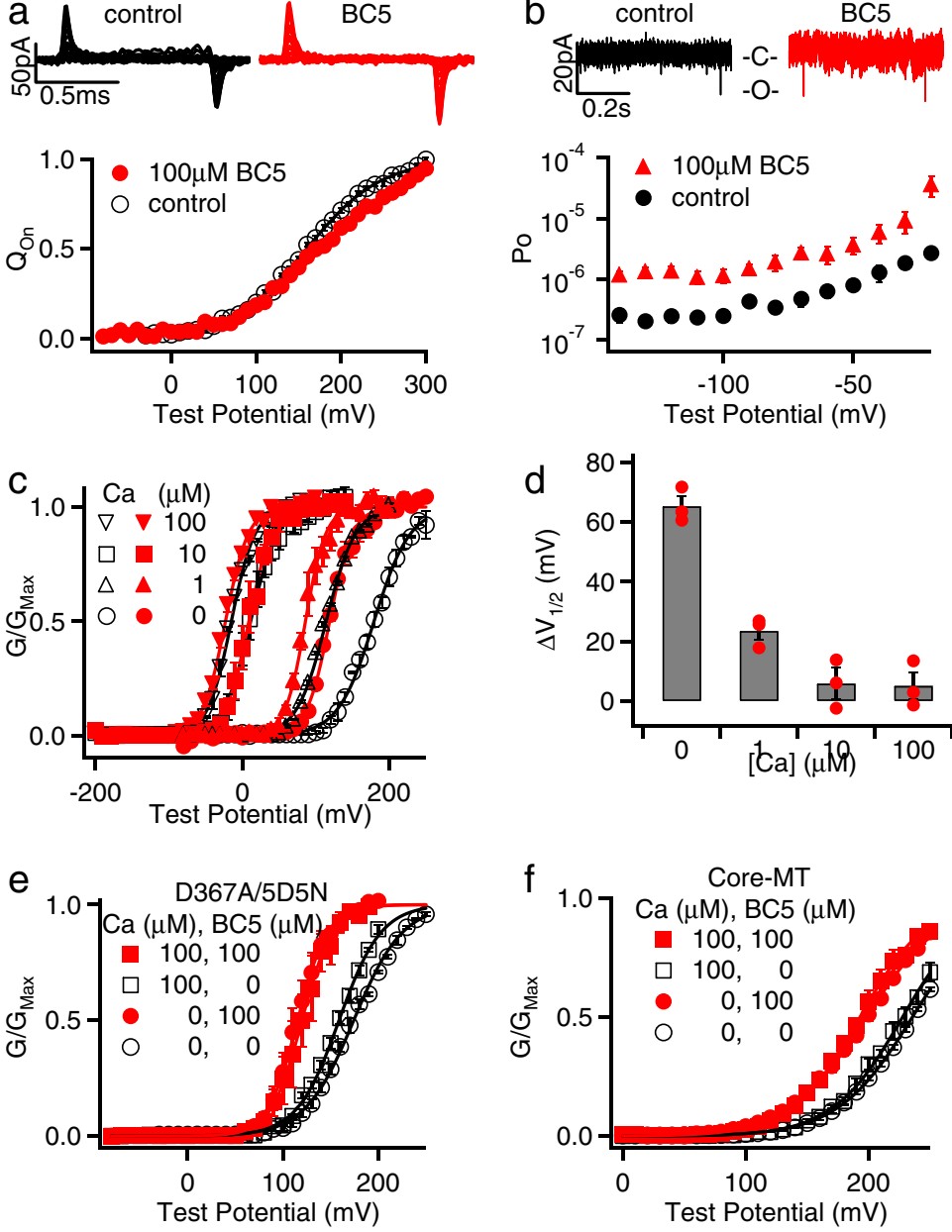

**Fig. 3 | BC5 perturbs Ca²⁺-dependent activation. a** Top, gating currents in control and 100 μM BC5. Voltage pulses were from −80 to 300 mV with 20 mV increments. Bottom, Normalized gating charge-voltage (QV) relation of on-gating currents. The smooth lines are fits to the Boltzmann function, $n = 14$ for control, and $n = 7$ for BC5. **b** Top, current traces of a patch containing hundreds of BK channels at −140 mV in control and 100 μM BC5. Only brief unitary channel openings were seen at the negative voltage. O: open; C: closed. Bottom, Open probability (Po) at negative voltages, $n = 9$ for control, and $n = 3$ for BC5. **c** GV relations in various [Ca²⁺]$_i$ (black hollow symbols) and in addition of 100 μM BC5 (red filled symbols). Black solid lines are Boltzmann fits without BC5 while red solid lines are Boltzmann fits with BC5.

$n = 3$ for all conditions. **d** GV shifts in response to 100 μM BC5 at different [Ca²⁺]$_i$. Data points are shown in solid circles, all of $n = 3$. **e** BC5 effects on the D367A5D5N mutant channel in 0 and 100 μM [Ca²⁺]$_i$. $n = 4$ for 0 [Ca²⁺]$_i$ control; $n = 5$ for 0 [Ca²⁺]$_i$ with 100 μM BC5; $n = 4$ for 100 μM [Ca²⁺]$_i$ control and $n = 3$ for 100 μM [Ca²⁺]$_i$ with 100 μM BC5. D367A and 5D5N (D897-901N) ablated the two Ca²⁺-binding sites in each Slo1 subunit, respectively. GV relations were fit with the Boltzmann function (solid lines). **f** BC5 effects on the Core-MT BK channel in 0 and 100 μM [Ca²⁺]$_i$. $n = 14$ for 0 [Ca²⁺]$_i$ control; $n = 9$ for 0 [Ca²⁺]$_i$ with 100 μM BC5; $n = 3$ for 100 μM [Ca²⁺]$_i$ control and $n = 6$ for 100 μM [Ca²⁺]$_i$ with 100 μM BC5. GV relations were fit with the Boltzmann function (solid lines).

by perturbing the pathway. Taken together, our results demonstrate the importance of the molecular interactions at the VSD-CTD interface of BK channels in coupling Ca²⁺ binding to pore opening. One of the two major allosteric pathways mediating Ca²⁺ binding in the CTD to open the channel gate in the membrane is via the VSD-CTD interface. It is not clear where the allosteric path eventually merges with that for voltage-dependent activation. However, since both Ca²⁺ and BC5 can activate the channel independently of voltage[13,23] (Fig. 3a, b) the allosteric path for Ca²⁺-dependent activation starting from Ca²⁺-

binding sites remains distinct from voltage-dependent activation at least down to the BC5 site.

## Discussion

We identified BC5 using in silico screening in order to modify Ca²⁺-dependent activation of BK channels (Fig. 1a, b). Our results showed that BC5 activated BK channels by selectively altering the Ca²⁺-dependent activation mechanism in the absence of voltage sensor activation and without indirectly modifying voltage-dependent gating

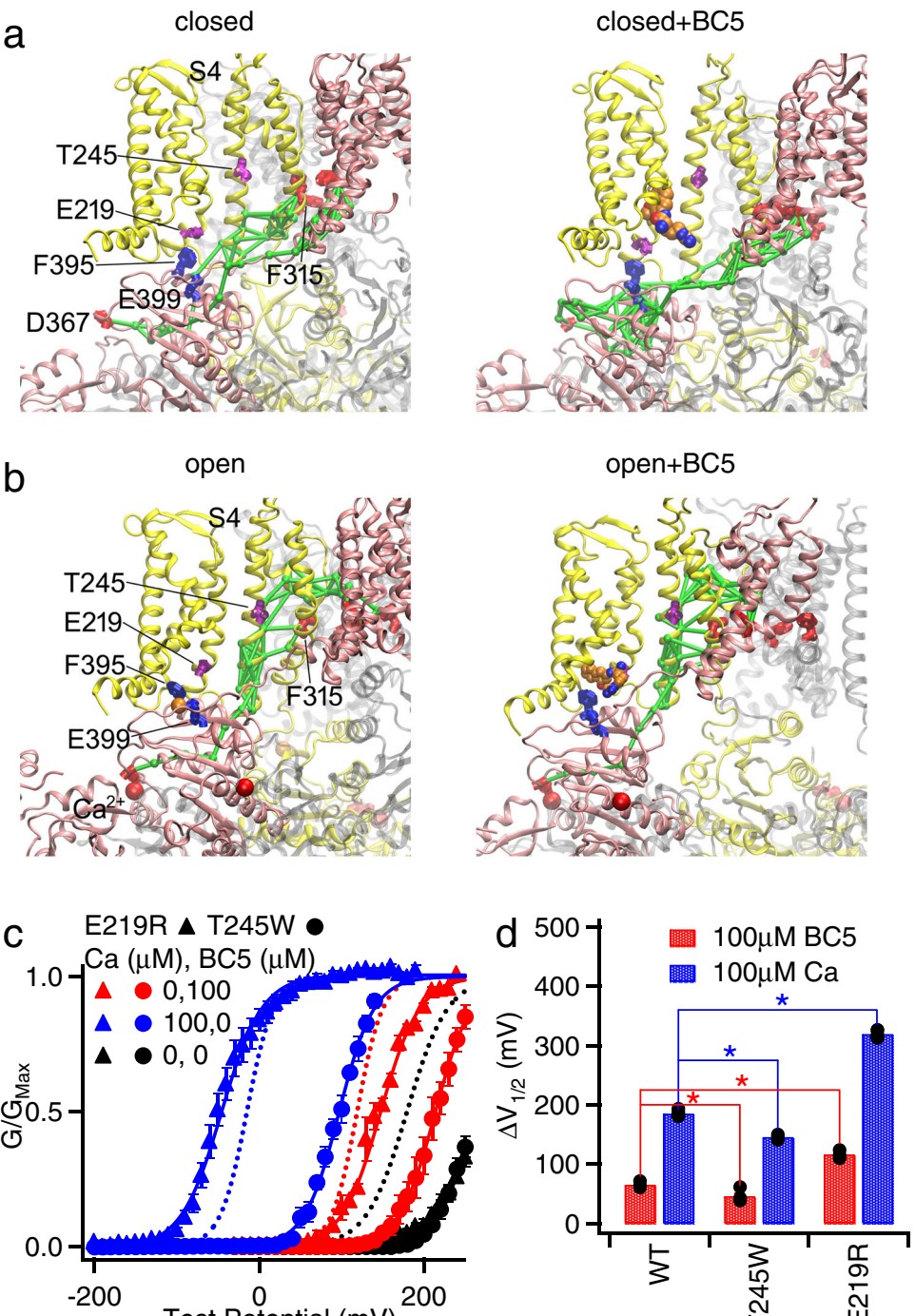

**Fig. 4 | BC5 interactions with the allosteric pathway for coupling Ca²⁺ binding to pore opening. a, b** Optimal and suboptimal pathways coupling Ca²⁺-binding site D367 and pore lining residue F315. A similar region as in Fig. 1a is shown. The two neighboring subunits forming the VSD-CTD contacts are colored in pink and yellow, respectively. The pathways are represented in green sticks. **c** GV relationship and Boltzmann fits (solid lines) of the mutant channels E219R (results are shown with triangles) and T245W (results are shown with circles). $V_{1/2}$ and slope factor (mV): for E219R, 263.7 ± 8.7 and 30.4 ± 6.8 at 0 [Ca²⁺]ᵢ and 0 BC5 (black triangles, $n = 5$); 147.4 ± 4.7 and 23.6 ± 4.1 at 0 [Ca²⁺]ᵢ and 100 μM BC5 (red triangles, $n = 3$); and -51.5 ± 4.2 mV and 24.7 ± 3.5 at 100 μM [Ca²⁺]ᵢ and 0 BC5 (blue triangles, $n = 6$); for

T245W, 260.9 ± 6.8 and 21.1 ± 5.3 at 0 [Ca²⁺]ᵢ and 0 BC5 (black circle, $n = 6$); 213.8 ± 2.9 and 22.1 ± 2.8 W at 0 [Ca²⁺]ᵢ and 100 μM BC5 (red circle, $n = 7$); and 106.0 ± 3.3 and 21.2 ± 3.1 at 100 μM [Ca²⁺]ᵢ and 0 BC5 (blue circle, $n = 3$). Dashed lines are taken from Fig. 3c for WT at 0 [Ca²⁺]ᵢ and 0 BC5 (black); at 0 [Ca²⁺]ᵢ and 100 μM BC5 (red); and at 100 μM [Ca²⁺]ᵢ and 0 BC5 (blue). **d** GV shift caused by 100 μM [Ca²⁺]ᵢ (blue, $n = 6$ for WT, $n = 3$ for T245W and $n = 6$ for E219R) and 100 μM BC5 (red, $n = 3$ for WT, $n = 4$ for T245W and $n = 3$ for E219R). Individual data points are shown in solid circles. Both mutations are significantly different in GV shift caused by 100 μM [Ca²⁺]ᵢ and 100 μM BC5, respectively, compared to WT with P<0.05, one-way Tukey–Kramer ANOVA test was used.

(Fig. 3a, b). Molecular docking, atomistic simulations and mutagenesis studies together suggest that BC5 binds at the CTD-VSD interface (Fig. 2), away from Ca²⁺-binding sites in the CTD and the pore in the membrane. Yet it not only promotes pore opening (Fig. 1d) but also

interacts with Ca²⁺ binding such that its effect on channel activation is inhibited by Ca²⁺ binding (Fig. 3c–f). Thus, BC5 perturbs both ends of the Ca²⁺-dependent activation mechanism, the sensor and the pore, allosterically by interacting with the pathway that couples Ca²⁺ binding

to pore opening. This pathway is built intrinsically in the BK channel protein, since BC5 still activates the Core-MT channel (Fig. 3f), in which the entire CTD is removed and replaced with an exogenous short peptide, indicating that the pathway downstream from the BC5-binding site is preserved despite the removal of $Ca^{2+}$ sensors.

We found that BC5 activated BK channels by shifting the GV relation (Fig. 2f) and increasing open probability at negative voltages (Fig. 3b) to a similar extent as 1 μM $Ca^{2+}$[47,52]. When the dependence of BK channel activation on $Ca^{2+}$ was fit with the MWC model, the $Ca^{2+}$-binding affinity at the open state was -1 μM[47,52]. Thus, BC5 activation of the channel only reached a part of the full capacity of $Ca^{2+}$-dependent activation (Fig. 4d). This is not considered surprising because $Ca^{2+}$ binding is coupled to pore opening via two major pathways, one is through the CTD-VSD interface, which is affected by BC5, and the other is through the C-linker[19–21,27,29,30,36,38] (Fig. 4a, b). However, in our dynamic network and information flow analyses, the highest information flow residues were observed at the VSD-CTD interface (Fig. 4a, b and Supplementary Fig. 9). These results suggest that the pathway through the VSD-CTD interface may make more contributions to $Ca^{2+}$-dependent activation than the C-linker. However, since the maximal activation by BC5 was less than half of that by $Ca^{2+}$ (Fig. 4d), BC5 interaction with the pathway may not be strong enough to even achieve the full capacity of $Ca^{2+}$-dependent activation via the CTD-VSD pathway. This idea is consistent with the result that $Ca^{2+}$ binding inhibited BC5 effects on channel activation (Fig. 3c, d), suggesting that the perturbation of the pathway by $Ca^{2+}$ binding was stronger and overrode that by BC5.

Each Slo1 subunit of BK channels contains two $Ca^{2+}$-binding sites in the CTD[41,47,53,54], and we found that $Ca^{2+}$ binding to both sites shared the same CTD-VSD allosteric pathway for coupling to the pore. We measured $Ca^{2+}$ effects on BC5 activation of the channel with either $Ca^{2+}$-binding site removed by mutation 5D5N or D367N[47]. BC5 activated the channel similarly to that in the WT BK in either of these mutant channels in the absence of $Ca^{2+}$. In 100 μM $Ca^{2+}$ BC5 activation was not abolished as in the WT channel (Fig. 3c, d and Supplementary Fig. 10), indicating that the disrupted $Ca^{2+}$ binding to either of the removed site by 5D5N or D367N could no longer inhibit BC5 effects via the CTD–VSD pathway. In other words, $Ca^{2+}$ binding to both sites is required to inhibit BC5 activation effects. These results indicate that both $Ca^{2+}$ binding sites are coupled to the activation gate via the same allosteric pathway that interacts with BC5. Since BC5 activated the 5D5N or D367N channels to a similar extent by a similar GV shift (Supplementary Fig. 10), the two $Ca^{2+}$-binding sites may activate the channel with a roughly equal amount via the allosteric pathway that interacts with BC5. Our dynamic pathway analysis shows that the part of the pathway downstream from the BC5 site on the VSD–CTD interface to the activation gate is common to both $Ca^{2+}$-binding sites, but the part of the pathway upstream from the BC5 site to the two $Ca^{2+}$-binding sites diverges to connect individual $Ca^{2+}$ sites (Supplementary Fig. 11).

BC5 activated BK channels as an allosteric agonist, but it also inhibited the channel as an off-target effect (Fig. 1c–e). Opposite to its activation of the channel, which was dependent on $Ca^{2+}$ (Fig. 3c, d) but not on voltage (Fig. 3a, b), the inhibition by BC5 was dependent on voltage but not $Ca^{2+}$ (Fig. 1f). While BC5 can be used as an excellent tool to study $Ca^{2+}$-dependent activation mechanisms, it can also be used as an effective BK channel inhibitor in physiological conditions, where the membrane voltage remains more positive than the $K^+$ equilibrium potential so that BK channels are always inhibited. Our results revealed the CTD-VSD interface as an excellent site for allosteric agonists (Fig. 2) and the inner pore as a site for inhibitors (Supplementary Fig. 4) of BK channels. It should be noted that our current study suggests that BC5 adopts multiple binding modes in the CTD-VSD interface and most contacts are transient. Thus, further chemical modification of BC5 may yield new compounds with more stable interactions with BK channels,

which may possess stronger allosteric activation activity. These results also suggest that potentially other sites along the allosteric pathways (Fig. 4a, b) can be used for allosteric agonists or antagonists to modulate BK channel function. These sites can be targeted for the development of therapeutic molecules to treat BK channel associated pathologies including drug therapies for BK channel associated diseases such as neurological disorders associated with either gain-of-function or loss-of-function BK mutations[7] or impaired urinary bladder control[11].

## Methods

### Molecular docking and in silico screening

A structure-based in silico screening strategy was employed in this study. This strategy was previously applied to the KCNQ1 channel, resulting in the discovery of the novel channel modulators[55,56]. Specifically, the in silico screening method used our molecular docking software MDock[57–60] as the docking engine, and each small molecule in a chemical library was docked to a selected site on a target protein. The protein structure was treated as a rigid body during docking and other parameters in the docking program were set as default. Then, compounds in the library were ranked by their binding scores evaluated by ITScore2 (implemented in MDock)[59], which is a knowledge-based atomic pairwise scoring function derived by using a statistical mechanics-based iterative method based on a training set of experimentally determined protein–ligand complex structures (a refined set of PDBbind 2012 containing 2897 protein–ligand complexes[61,62]).

As for the protein structure for docking, both open ($Ca^{2+}$-bound, PDB: 6v38) and closed ($Ca^{2+}$-free, PDB: 6v3g) conformations of human BK channels were solved using Cryo-EM[21]. A major difference between these two conformations is the relative position between the membrane spanning region (S0-S6) and the cytosolic domain (CTD), as shown in Supplementary Fig. 1a. Because we aimed to discover chemical compounds stabilizing BK channels during their open state, the interface between the voltage sensor domain (S1-S4) and CTD was selected for the target site and the structure in the open state (PDB: 6v38) was used in the in silico screening study (see Fig. 1a and Supplementary Fig. 1b). Because the residues interacting with $Mg^{2+}$ are located at the VSD-CTD interface and were included in our target site, the $Mg^{2+}$ ion in the channel structure was removed before docking. The protein structure was prepared using UCSF Chimera[63,64]. Moreover, the experimental hit, BC5, was docked to both open and closed conformations of BK channels. The resulting binding modes were used as starting points for atomistic simulations.

Next, to prepare for the chemical library in this proof-of-concept study, considering that the VSD-CTD interface contains many charged residues (e.g., K98, D99, D173, and E219 on VSD, and R342, E374, K392, and E399 on CTD), we screened a subset of the Available Chemical Database (ACD version 1999, Molecular Design Ltd., currently is a part of BIOVIA in Dassault Systèmes, https://www.3ds.com/) in which each compound contains either one charged group (carrying a charge of +1, −1, +2 or −2) or two charged groups (with one group carrying +1 charge and one group carrying −1 charge). There were a total of -4 × 10⁴ compounds in this chemical sub-database for screening. Because accurately predicting protein–ligand binding energies remains a challenging task for scoring functions in molecular docking programs[64,65], current in silico screening methods are in pursuit of an enriched subset of potential candidates[66,67]. Therefore, we further visually inspected the docking modes of the top 2% compounds ranked by the scoring function, ITScore2[59]. Compounds forming strong interactions (e.g., hydrogen bonds or salt bridges) with both VSD and CTD in their docking modes were selected. One of the selected compounds, BC5, forms hydrogen bonds with N172 and D173 from VSD and S337 and E399 from CTD, respectively. At the end, 9 purchasable compounds (reported in Supplementary Table 3) were selected for experimental assays.

## Mutations and expression

All mutations in this study were made by using overlap-extension polymerase chain reaction (PCR) with Pfu polymerase (Stratagene) on the template of the mbr5 splice variant of *mslo1* (Uniprot ID: Q08460)[68]. Sequencing of the PCR-amplified regions is used to confirm the mutation[41]. mRNA was synthesized in vitro from linearized cDNA using T3 polymerase kits (Ambion, Austin, TX). mRNA of mutations was injected into oocytes (stage IV–V) from female *Xenopus laevis* at an amount of 0.05–50 ng/oocyte. The injected oocytes were incubated at 18 °C for 2–7 days before electrophysiology recordings. We have complied with all relevant ethical regulations and animal protocol is approved by IACUC at Washington University in St. Louis, USA.

## Electrophysiology

All experimental data were collected from inside-out patches. The set-up included an Axopatch 200-B patch-clamp amplifier (Molecular Devices, Sunnyvale, CA), ITC-18 interface and Pulse acquisition software (HEKA Electronik, Lambrecht, Germany). Borosilicate pipettes used for inside-out patches were pulled using a Sutter P-1000 (Sutter Instrument, Novato, CA) and then wax coated and fire polished with a resistance of 0.5–1.5 MΩ.

The macroscopic currents were collected at the 50 kHz sampling rate (20-μs intervals) with low-pass-filtered at 10 kHz. A P/4 leak subtraction protocol with a holding potential of –120 mV was applied to remove capacitive transients and leak currents (except for experiments in Fig. 3b). The solutions used in recording macroscopic ionic currents include: (1) Pipette solution (in mM): 140 potassium methanesulfonic acid, 20 HEPES, 2 KCl, 2 MgCl$_2$, pH 7.2. (2) The nominal 0 μM [Ca$^{2+}$]$_i$ solution contained about 0.5 nM free [Ca$^{2+}$]$_i$ (in mM): 140 Potassium methanesulfonate, 20 HEPES, 2 KCl, 5 EGTA, pH 7.1. (3) Basal bath (intracellular) solution to make different [Ca$^{2+}$]$_i$ (in mM): 140 Potassium methanesulfonate, 20 HEPES, 2 KCl, 1 EGTA, and 22 mg/L 18C6TA, pH 7.2. CaCl$_2$ was then added into the basal solution to obtain the desired free [Ca$^{2+}$]$_i$ that was measured by a Ca$^{2+}$-sensitive electrode (Thermo Electron, Beverly, MA).

Gating currents (Fig. 3a) were recorded from inside-out patches at the 200 kHz sampling rate and 20 kHz filtering with the same leak subtraction protocol. (4) Pipette solution for gating currents (in mM): 125 tetraethylammonium (TEA) methanesulfonic, and 2 TEA chloride, 2 MgCl$_2$, 20 HEPES, pH 7.2 and (5) bath solution for gating currents (in mM): 135 N-methyl-D-glucamine (NMDG) methanesulfonic, 6 NMDG chloride, 20 HEPES, 5 EGTA, pH 7.2.

Single-channel currents (Fig. 1g, h) were collected using an Axopatch 200B and sampled at 200 kHz with a Digidata 1322A (Molecular Devices). Filtering for analysis was at 5 kHz[69].

BC5 (ARG-4-methoxy-2-naphthylamine, ordered from Medchemexpress LLC Monmouth Junction, NJ) was dissolved into DMSO to make a 100 mM stock solution and then added to recording solutions to achieve the indicated concentrations. All other chemicals were purchased from Sigma-Aldrich. All the experiments were performed at room temperature (22–24 °C).

## Data analysis

Conductance–voltage (GV) relationship was determined by measuring macroscopic tail current amplitudes at –80 or –120 mV. The GV relationship was normalized to the maximum G value (G$_{Max}$) and was fitted with the Boltzmann function:

$$G/G_{Max} = 1/(1 + \exp(-ze_o(V - V_{1/2})/kT)) = 1/(1 + \exp((V_{1/2} - V)/b)) \tag{1}$$

where $z$ is the number of equivalent charges that move across the membrane electric field, $e_o$ is the elementary charge, $V$ is membrane potential, $V_{1/2}$ is the voltage where $G/G_{Max}$ reaches 0.5, $k$ is Boltzmann's constant, $T$ is absolute temperature, and $b$ is the slope factor with units

of mV. Each GV relationship presented in figures was the averaged results of 3–9 patches and error bars in the figures indicate the standard error of mean (SEM).

In order to measure limiting slope the open probability (P$_O$) and the total open probability for all the channels N in a patch (NP$_O$) at negative voltages (Fig. 3b) were first measured. The currents of all opening events from a patch containing hundreds or thousands of channels during a long pulse (5 or 10 s) at each voltage were integrated, and then divided by the single channel current amplitude and total time to obtain NP$_O$.

The estimation of the total number of channels expressed in a membrane is from the equation

$$I = N\gamma P_O(V - E_K) = Ni \tag{2}$$

where $i$ represents single channel current and $i = P_O(V - E_K)$. $N$ was estimated by measuring the macroscopic current at 100 mV in the presence of 100 μM [Ca$^{2+}$]$_i$, where the single channel conductance is 273 pS and $P_O$ is about 1.0.

Dose response curve of G–V shifts ($\Delta V_{1/2}$) or current inhibition on BC5 concentration (EC$_{50}$ in Figs. 1e, 2e, f, IC$_{50}$ in Fig. 1e, and Supplementary Fig. 4b) was obtained from the fitting to the Hill equation:

$$\theta = \Delta V_{1/2}^{max}/[1 + (EC_{50}/X)^n] \tag{3}$$

Where $\theta$ is the expected response at dosage $X$, $\triangle V_{1/2}^{max}$ is the maximum response for an infinite dosage, EC$_{50}$ is the dosage at which the response is 50% of maximum, and $n$ is the Hill Coefficient. In all of our final fittings Hill Coefficient was set to 1 since it was close to 1 during the initial fits. The curve was fitted using the "curve_fit" fuction Python Scipy module. The standard errors of EC$_{50}$ and $\triangle V_{1/2}^{max}$ were estimated directly from the SEMs of the input data and validated by numerical simulations.

## Reproducibility

The number of experiments (patches) n for some figure panels are listed below. Figure 1d, $n = 12$ for control, $n = 3$ for 0.01 and 0.03 μM BC5; $n = 10$ for 0.1 μM BC5; $n = 4$ for 0.3 and 1 μM BC5; $n = 5$ for 2 μM BC5; $n = 6$ for 5 μM BC5; $n = 3$ for 10 μM BC5; $n = 5$ for 30 and 100 μM BC5; $n = 4$ for 300 μM BC5.

Figure 1e, for GV shift, $n$ is the same as Fig. 1d; For current inhibition by BC5, $n = 12$ for control, $n = 4$ for 0.01, 0.1 and 0.3 μM BC5; $n = 5$ for 1 and 2 μM BC5; $n = 4$ for 5 μM BC5; $n = 3$ for 10 μM BC5; $n = 5$ for 30 and 100 μM BC5; $n = 4$ for 300 μM BC5;

Figure 2e, for WT, $n$ is the same as shown in Fig. 1d; for I105E, $n = 4$; for D173C, $n = 5$; for E180A, $n = 6$; for N182A, $n = 8$; for E395A, $n = 5$; for E399R, $n = 3$; for DM1 (E180AN182A), $n = 3$; for DM2 (I105EE399R), $n = 3$.

## Atomistic simulations

The cryo-EM structures of hBK channel in the Ca$^{2+}$-bound (PDB: 6v38) or Ca$^{2+}$-free (PDB: 6v3g) states were used in all simulations reported in this work[21]. Several short loops absent only in one structure were rebuilt using the other structure as template using the Swiss-PDB server[70]. For example, A567-E576 and N585-E591 missing in Ca$^{2+}$-bound structure were rebuilt using the Ca$^{2+}$-free structure. Missing N- and C-terminal segments as well as several long loops (e.g., V53-G92, A614-V683, and D834-I870) are presumably dynamic and thus not included in the current simulations. Residues before and after the missing segments are capped with either an acetyl group (for N-terminus) or a N-methyl amide (for C-terminus). Standard protonation states under neutral pH were assigned for all titratable residues.

As summarized in Supplementary Table 1, four sets of atomistic simulations were performed: The Ca$^{2+}$-bound state of BK channel with/without BC5 (*sim 1-3* and *4-6*, respectively) and the Ca$^{2+}$-free state of BK channel with/without BC5 (*sim 7-9* and *10-12*, respectively). The initial binding pose of BC5 was identified with docking study. In

simulations *1-3* and *6-9*, a BC5 molecule is placed in binding pocket in each of the four subunits.

All initial structures were first inserted in model POPC lipid bilayers and then solvated in TIP3P water using the CHARMM-GUI web server[71]. The systems were neutralized and 150 mM KCl added. The final simulation boxes contain about ~900 lipid molecules and ~130,000 water molecules and other solutes, with a total of ~680,000 atoms and dimensions of ~185 × 185 × 160 Å³. The CHARMM36m all-atom force field[72] and the CHARMM36 lipid force field[73] were used. The interaction parameters of BC5 were then assigned by identifying similar atom, bond, angle, and dihedral-angle types from similar small molecules within the CHARMM36m all-atom force field and CHARMM General Force Field (Charmn36cgenff)[72] (see Supplementary Fig. 12). All simulations were performed using CUDA-enabled versions of Gromacs 2018[74,75]. The default CHARMM-GUI generated configuration was used except otherwise noted here. Electrostatic interactions were described by using the Particle Mesh Ewald (PME) algorithm[76] with a cutoff of 12 Å. Van der Waals interactions were cutoff at 12 Å with a smooth switching function starting at 10 Å. Covalent bonds to hydrogen atoms were constrained by the SHAKE algorithm[77], and the MD time step was set at 2 fs. The temperature was maintained at 298 K using the Nose-Hoover thermostat[78,79]. The pressure was maintained semi-isotropically at 1 bar at membrane lateral directions using the Parrinello–Rahman barostat algorithm[80].

All systems were first minimized for 5000 steps using the steepest descent algorithm, followed by a series of equilibration steps where the positions of heavy atoms of the protein, BC5 and lipid were harmonically restrained per CHARMM-GUI Membrane Builder protocol[71]. Briefly, 6 equilibration step (25 ps for steps 1–3, 100 ps for steps 4–5 and 10 ns for step 6) were performed, where the restrained force constant for proteins and BC5 molecules were set to 10, 5, 2.5, 1.0, 0.5 and 0.1 kcal mol$^{-1}$Å$^{-2}$, respectively. For lipids, the phosphorus is restrained with force constants of 2.5, 2.5, 1.0 and 0.5, 0.1 and 0.0 kcal mol$^{-1}$Å$^{-2}$, respectively. In the last equilibration step, only protein heavy atoms were harmonically restrained and the system was equilibrated for 10 ns under NPT (constant particle number, pressure and temperature) conditions at 298 K and 1 bar. All production simulations were performed under NPT conditions at 298 K and 1 bar.

### Analysis

Unless stated otherwise, snapshots were extracted every 50 ps for all equilibrium MD trajectories for calculation of statistical distributions. Molecular illustrations were prepared using VMD[81]. Contact probability was calculated with MDAnalysis[82]. Specifically, a residue is considered to have contact with BC5 if any of its non-hydrogen atoms is found within 6 Å of any heavy atoms of BC5. The results were first averaged over the four BC5 in four BK subunits, then averaged over three simulations in each condition.

### Dynamic network analysis

The analysis identifies the most probable pathways of dynamic coupling between two selected residues using the Floyd-War-shall algorithm[49]. For this, each residue (and BC5) is represented as a node of the network. If two residues have a contact (identified with a minimal non-hydrogen atom distance cutoff of 5 Å) for greater than 75% of the simulation time, an edge connecting the two corresponding nodes is added to the network. The resulting contact matrix was weighted based on the covariance of dynamic fluctuation ($C_{ij}$) calculated from the same MD trajectory as $w_{ij} = -\log(|C_{ij}|)$. The length of a possible pathway $D_{ij}$ between distant nodes $i$ and $j$ is defined as the sum of the edge weights between consecutive nodes along this path. The optimal pathway is identified as the shortest path, thus having the strongest dynamic coupling. Suboptimal paths between $i$ and $j$ are identified as additional top paths ranked using the path length. The analysis was performed using the Network View[83]. The first 20 optimal and suboptimal paths were selected for analysis presented in this work.

### Information flow

Information flow provides a global assessment of the contributions of all nodes to the dynamic coupling between selected "source" and "sink" nodes[50,51]. This analysis complements the dynamic pathway analysis to provide additional insights on how different residues may contribute to sensor–pore coupling. For this, we first construct a network similar to the one described above. Pairwise mutual information was calculated between node $i$ and $j$ as follows: $M_{ij} = H_i + H_j - H_{ij}$. $H_i$ is calculated as $\frac{1}{N}\sum_{n=1}^{N}[-\ln\rho_i(x)]$, where $\rho_i(x)$ is the fluctuation density and $x$ is the distance to the equilibrium position[51]. Gaussian mixture model (GMM)[84] is used to estimate the density. The residue network is then defined as $A_{ij} = C_{ij} M_{ij}$, in which $C_{ij}$ is the contact map. To analysis the information flow from source ($S_0$) to sink ($S_l$) nodes, the network Laplacian, is calculated as $L = D - A$, where $D$ is diagonal degree matrix: $D_{ii} = \sum_j A_{ij}$. The information flow through a given node (residue) is defined as $f_i = \frac{1}{2}(\sum_j|(P_i - P_j)|A_{ij})$. The potentials $P$ is given by $P = \widetilde{L}^{-1}b$, where $\widetilde{L}^{-1}$ is inverse reduced Laplacian and $b$ is b is the supply vector which corresponds to one unit of current entering at the source node that will exit at sink nodes. The magnitude of $f_i$ thus quantifies the contribution of residue $i$ to dynamic coupling between the source and sink nodes.

### Reporting summary

Further information on research design is available in the Nature Research Reporting Summary linked to this article.

## Data availability

The data that support this study are available from the corresponding authors upon reasonable request. Previously published accession codes used in this study: 6V38 and 6V3G. Source data underlying Figs. 1–4 and Supplementary Figs. 3–5, 9, and 10 are available as a Source data file. Source data are provided with this paper.

## Code availability

Custom code used in this study is publicly available at GitHub (https://github.com/zhiguangjia/Ananlysis-code-for-BC5-Nature-communication-).

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

## Acknowledgements

This work was supported by grants R01 HL142301 (J. Cui and J.R.S.), R35 GM136409 (X.Z.), R35 GM144045 (J. Chen), and R01 GM114694 (K.L.M.) from NIH.

## Author contributions

J. Cui, X.Z., and J. Chen conceived and initiated the studies. J. Cui, X.Z., J. Chen, J.R.S., and K.L.M. provided supervision and management. G.Z., H.L., J.S., M.M., C.A., J.R.S., and J. Cui performed electrophysiological experiments and analyzed data. X.X. and X.Z. performed molecular modeling and in silico screening. Z.J. and J. Chen performed molecular dynamic (MD) simulations, dynamic network and information flow analyses. Y.G. and K.L.M. did experiments and analyzed data on single channels. G.Z., Z.J., J. Chen, X.X, X.Z., and J. Cui wrote the manuscript, with input from K.L.M, J.R.S., and C.A.

## Competing interests

The authors declare no competing interests.
