## [Peer Review File · Nature Communications]

An allosteric modulator activates BK channels by perturbing coupling between Ca²⁺ binding and pore openingReviewers' Comments:

Reviewer #1:

Remarks to the Author:

This work by Zhang et al explores the mechanisms underlying allosteric modulation of pore opening in BK channels, which remains a key yet unanswered question in the field. Several findings in recent years, including the elucidation of Ca²⁺-bound and Ca²⁺-unbound structures and other functional studies (well recognized in the manuscript), point to a relevant role of the CTD-VSD interface in these mechanisms. Specifically, Zhang et al identify a novel molecule by in silico screening, BC5, that interacts dually with the channel, producing two functional effects: blocking of outward currents, possibly by interaction with the pore, and activation of tail currents, by interacting with the VSD-CTD interface, at a location coincident to great extent with the regulatory Mg²⁺ binding site. Consistent with this fact, the authors demonstrate that BC5 partially competes with Mg²⁺ for the binding site, and mutations abolishing Mg²⁺ activation also partially abolish BC5 activation effects. This is further confirmed by MD and site mutation analyses (but see some comments below). Quantification of gating currents shows that VSD function is not altered by BC5, whereas intrinsic opening is increased, leading the authors to conclude that even though they bind at close locations, the BC5 activation effect involves a mechanism different from that of Mg²⁺, specifically by modulating the Ca²⁺-dependent pathway. Further experiments are performed using different Ca²⁺ concentrations and BK constructs with impaired Ca²⁺ binding, to propose that BC5 may interact with the Ca²⁺-binding-to-pore-opening molecular pathway, which is also supported by specific mutations of residues potentially involved. The work is solid, experiments are well planned and performed and the manuscript is very well written and structured. The topic is highly timely and provides interesting ideas, setting the background to test intriguing new hypothesis related to BK function and, more generally, allosteric protein function. I believe that it will be of great interest to the extended scientific community. Some concerns, comments and suggestions are listed below.

Line 155 and Fig. 2: Regarding the results with mutant E99R, the authors show that the V-half shift produced by BC5 is reduced (although not to a great extent) by the mutation, consistent with the idea that BC5 and Mg²⁺ bind to at least partially common sites. Does the inhibitory effect of BC5 remain the same in this mutant? If this is the case, this would further support the idea proposed by the authors that the inhibitory and activation effects of BC5 on BK function are unrelated.

The authors mention that the E399 residue is also shared by both Mg²⁺ and BC5 binding sites (line 150). This residue is mutated to R as shown in Fig. 2f, with apparently very different effects than the potentially analogous mutation of D99 to R shown in Fig. 2c. It is true that EC₅₀ is increased, but the V-half shift seems to be increased as well (this also applies to DM2 containing the same mutation. Data related to mutant D99R are not included in Fig. 2f so comparison is not possible. Can this be better clarified?

Figure 2f: what is the rationale for double mutations DM1 and DM2?

Figure 2a: This representation may mislead some unaware reader. Although it is understood that Mg²⁺ and BC5 cannot bind simultaneously to the binding site, perhaps a different representation could be used to make this aspect perfectly clear, if this is the case.

Line 153: "BC5 at the same concentration". Not clear what concentration is this referring to, same as Mg²⁺? What is the criterion to use this concentration for comparison with Mg²⁺ effect?

The authors propose that the interaction site is very similar to that of Mg²⁺ regulating the channel, however they function in a radically different way. The difference between underlying mechanisms is merely described in the text but I find that this aspect could be further discussed.

Fig. 2f: No statistical data is provided here. would it be possible to add eg the confidence intervals?

Regarding the dynamic network and information flow analyses (Fig. 4, line 214 and beyond): This analysis does not include the RCK2 site. Is this assuming that the activation pathway from Ca²⁺ binding to RCK2 is common to, and/or necessarily includes activation of the RCK1 site? How does this relate to the findings of Lorenzo-Ceballos et al (Ref 33 in the manuscript)?

Text should be revised for typographical errors. Some examples:

Line 25: spaniing
Line 31: mutagenesis
Line 269: patheay

Reviewer #2:

Remarks to the Author:

The ms by Zhang et al addresses the mechanism of a novel BK channel activator, BC5, identified from a relatively small subset of compounds (40K) by an in silico screen that targeted the VSD-CTD interface. BC5 was observed to elicit a shift in the G-V curve toward negative voltages, determined from tail current measurements. At the same time, BC5 was observed to strongly block outward current at positive voltages. The activating and blocking effects may arise from separate mechanisms, because mutations targeting the presumed binding site at the VSD-CTD interface can reduce the activating effect, while the blocking effect remains intact. BC5 effectively activates channels in the absence of Ca, while BC5 appears to compete with Ca activation. The authors conclude that the VSD-CTD interface is critical for transducing the Ca-activation mechanism from the CTD to the pore.

Overall the ms is generally well written and the experimental data are of high quality. The discovery of a new pharmacological probe of BK channels targeted to the VSD-CTD interface will be of interest. There are, however, some concerns that should be addressed in the interest of improving the presentation.

1) It would be worthwhile to present some more details of the in silico screen. Prior to screening, was the target binding pocket identified arbitrarily, or was a computational approach used for this as well? Also, why was the screening space limited to charged molecules?

2) It is not entirely clear that the activating effect of BC5 is completely separate from the apparent pore-blocking effect. Isn't it possible the BC5 could be binding in the pore and stabilizing the open state, thus slowing the deactivation kinetics (as shown in Fig S1)? Are single channel open times at negative voltages increased in the presence of BC5? Although P_o is shown in Fig 3, it is not clear whether the open times are longer. If so, this would further support the idea that the drug is activating the channel under conditions where it is not blocked.

3) I'm not sure whether it is appropriate to call BC5 an activator/agonist, because the net effect of this compound under physiological conditions is to block outward current, and the blocking effect is actually stronger than the activating effect (with a lower IC_{50}). The authors may wish to re-word the title and take out the term "allosteric agonist".

Reviewer #3:

Remarks to the Author:

The paper presents the identification of a single compound which alters BK channel physiological parameters in an interesting way. The introduction is unclear and there are important pieces of information missing from the methodological section, affecting the reproducibility of the results and suggesting that the binding pocket could have been wrongly identified. The experimental procedures could have been carried further to inquire on the pharmacological properties. The importance of the finding is not fully explained in the therapeutic setting. Major improvements are needed to raise the manuscript to the standards of the journal, to allow full result reproducibility and to put the discovery in perspective.

Major comments

-The introduction lacks a logical flow. The second paragraph from the introduction starting at line 58 is very confusing to the reader. This paragraph discusses the known mechanisms of activation and should be placed after the BK channel elements (especially the sensors) have been put into structural context. The paragraph must present in a more clear and concise way the two sensors. Take for instance the phrase "In these activation mechanisms Ca²⁺ or voltage alters the conformation of its sensor" what sensor are we talking about? Ca²⁺ and voltage at once or each other individually are able to "alter" the sensor(s)? Next phrase "although Ca²⁺ and voltage activate BK channels via distinct mechanisms, the two activation mechanisms influence each other that can be described with an allosteric linkage" What are the two activation mechanisms? Please introduce them very clearly after the structural explanation. Then explain the intertwining between them (e.g. Ca-activation is independent of the VSD state).

-The third paragraph of the introduction should be split in two, as it is too large. As mentioned, one paragraph should be devoted to the structural features of the BK channel. Such explanation was attempted in this third paragraph. The second part of the third paragraph, as I see, is more focused on the detailed intricacies of channel activation and should be placed after the general description of how the two sensors/activation mechanisms work. The clarity of these ideas must follow, with the reader in mind.

-The Mg-based activation is firstly introduced very late in text, in line 93. There are no previous references to the Mg-activation possibilities manifested by BK, nor an explanation of the binding sites of Mg⁺⁺ vs Ca⁺⁺.

-All the structural features presented must be accompanied by placement of flanking or important amino acids (residue numbers, first and last of various sections of the protein) and by a small Figure overviewing the main structural elements of the channel, in both monomeric and multimeric forms. Such a Figure may include both 3D renderings and a 2D membrane-spanning view and could be included in Figure 1. Multiple references in the text could be made to this figure.

-throughout the body of the paper there is not an instance mentioning the two regulators of potassium conductance (RCK), parts of the cytosolic domain. Yet, at figure S8, the mutation D367N is said to ablate the Ca-binding site in RCK1.

-the molecular docking Method is poorly reproducible (to the extent of the impossibility of reproduction). There is no fixed mandatory word limit for the Methods section in Nat Commun (the informal one is 3000 words but this can be exceeded and the section in the paper spans about 2000 words) thus there is room for improvement. Has the Mdock piece of software ever been used in other similar virtual screening situations? Please provide references if so. If not, please state why it is optimal for such purpose. Does Mdock have any settings that can be tweaked by the user? What parameters were set for docking algorithms? Overall, the virtual screening is a little too straightforward and this is not acceptable, particularly given limitations of in silico approaches as compared to experimental screens (please compare the process of virtual screening to the one mentioned in various papers e.g. Chemical Physics Impact, Volume 3, December 2021, 100048) and could have been accompanied by a larger experimental screen (of more than 9 compounds).

-What is the "Available Chemical Database" (ACD, Molecular Design Ltd.)? I cannot seem to find it anywhere, nor can I find information about the company. Please provide links and mentions as to how any researcher can reach this Database. Has it been used in other works from the Authors or other researchers? If yes please cite accordingly. Why was this Database selected (I assume it is not free) and not freely available ones derived from renowned PubChem or ZINC (time-proven, with thousands of citations)? Does it possess any advantage? Does it cover the diversity of the chemical space accordingly?

-Was the flexibility of the protein residues accounted for in the virtual screening?

-What program was used to handle the PDB file(s)?

-What is the rationale for selecting PDB version 6V38 (Ca-bound) and not other states (e.g. Ca-free) for docking?

-Where was the searching space located? Page 6 line 123 states is to be "a site near the VSD-CTD interface" but this is very insufficient for reproducibility. What was the center and the radius of the searching space (provided it was a sphere, but this is not mentioned)? What is the rationale for removing the Mg ion from the PDB file? Line 148 at page 7 states that "docking site of BC5 in BK channels is located near the Mg²⁺ binding site", so does not this mean that it does not overlap with the Mg-binding site? Is it just "near" or is the docking site overlapping with the Mg-binding site? Should the ligand prevent Mg binding?

-What is the rationale for selecting a subset with charged compounds? Is the subset formed solely based on this criterion (charge +1 or +2) or were there other criteria for selecting the subset? How many compounds were there in the initial set?

-In which sense were the top candidates "manually inspected" (I assume "visually" would be a better choice of words)? Was the virtual screening carried out multiple times to ensure a statistical analysis of variability? Exactly how many compounds were in the top 2% of the ranked candidates? I assume all the compounds were ranked. This leads to 2% of the 40000 compounds, which is about 800 compounds, from which only 9 were "manually" chosen? The selection of 9 compounds out of 800 is perhaps the most important substep and must be accurately described. Nonetheless, it does not seem feasible nor practical. A supplementary table or figure of the process of selecting the compounds is needed (such as a workflow or pipeline etc.), which may include the predicted binding free energies from Mdock. Based on what criteria were the 9 candidates selected?

-Are all compounds in the subset commercially available? What can be stated of the other eight compounds, about their names, structures and computational or experimental features? The docking-based screen is the first and foremost step of the methodology, yet it is very poorly described and BC5 seems to be a serendipitous result. If only 1 one of the 9 candidates was positive for your interest, why would the other 800 or so not be eligible for even greater impact?

-How exactly were the interaction parameters assigned to the ligand for MD simulations? What tool was used to do this tedious task? What "similar small molecules" from the CHARMM36m force field were selected? The CHARMM36m is a force field for proteins and other tools have to be used, such as CGenFF or SwissParam etc. Parameterizing a small molecule for MD simulations is a very complex process and prone to errors, which is why automated tools combined with (if needed) manual adjustments are needed.

-Also, was the small molecule manipulated in any way in some software? In what format were its coordinates? Was it built from scratch or it was in pdb format from the start? Did CHARMM-GUI accept it as an input without any modifications?

-CHARMM-GUI already provides simulation parameters (input configuration files). Please provide more clear information on what was left at default and what was changed and why.

-There is insufficient proof to ascertain that the binding site is actually correctly identified. The ligand "wobbles" around during MD simulations and has zero stability. The docking method is used to identify stable binding poses of ligands; the MD actually shows this is not the case thus defeating the whole purpose of docking. Basically, the MD does not add new insights to the paper as it only proves the ligand "stays" in a presumed binding pocket. It is well known that stability of the ligand during MD simulations is indicative of its effect on the binding site and protein (J Comput Aided Mol Des 2017

Feb;31(2):201-211). The ligand binding is described simply and thus poorly in plain terms of probability of residue contacts, with poor mentions whatsoever about actual molecular interactions (e.g. H-bonds, pi-stacking etc.) although judging from the presumably high RMSD during simulations all the bonds seem transient. Also, the effects of the ligand are manifested on a channel with natural allosteric modulation, which makes it even harder to accurately pinpoint a binding site for a ligand. In my opinion, this is a major limitation that should be explicitly mentioned in the Discussion.

-In line with the previous comment, page 12 line 262 states that "BC5 binds at the CTD-VSD interface"; this may only be certainly demonstrated using a direct physical method such as crystallography, NMR spectroscopy etc. Please use words such as "suggest", "putative" or "indicate". Throughout the paper there should be more skepticism about this binding site and the results from MD should be treated as "indicative"; furthermore, it cannot be excluded that the ligand may bind at multiple sites at once!

-There is no mention of how the eight mutations included in the text and shown in Figures 2e and 2f alter the WT physiological parameters in the absence of the ligand. Are they completely neutral in this regard? This may actually emphasize that the binding site was more correctly identified. Also related, mutation N182A alters agonist's effects similar to the other five tested single residue replacements, yet this residue does not contact the ligand directly in docking or MD. Please provide explanations for this effect.

-Please clearly emphasize the importance of the identified compound as compared to other agonists (Molecular Pharmacology December 2, 2020, MOLPHARM-AR-2020-000106; Molecular Pharmacology April 30, 2022, MOLPHARM-AR-2021-000478; 3552-3557; PNAS February 28, 2012 vol. 109 no. 9) in a paragraph in Discussion.

-The discussion should be more focused on the novel discoveries made by the Authors, delineating the elements which have not been known previously from other papers.

-The forte of the paper is not sufficiently emphasized, that of discovering a ligand that has novel properties compared to other ligands. This is what needs to be put into therapeutical context at the end of the paper. There is no ADMET data/prediction about this compound, nor any mentions about its use in other contexts, but could it be useful as a lead compound? Its extensions as a lead compound for future studies could be mentioned, similarly to the recent experimental screening of Srinivasan et al. (Molecular Pharmacology April 30, 2022, MOLPHARM-AR-2021-000478).

Minor comments

-there are two words spelled wrong in the Abstract: spaniing and mutagensis

-the abbreviation CTD stands for "cytosolic tail domain" (not just "cytosolic domain"), please correct where needed

-page 4 line 73 "fundamental" instead of "foundamental"

-page 3 line 45 "On the other hand" implies that a previous phrase starts with "On one hand" please correct. Same on line 68. Same on line 82.

-page 4 line 61 "alter" instead of "alters"

-page 5 line 90 "tugging" should be placed between apostrophes (quotation marks or similar)

-line 316 the charges cannot be neutral, are either positive +1 or negative -1

-line 269 "pathway"

-line 273 "extent"

-Figure 2d there is a thick line ligand drawn in the Figure but it is not mentioned in the legend (only the thin lines should be depicted)

-Line 252 states "It is not clear whether the allosteric path eventually merges with that for voltage dependent activation" while line 68 states "the paths for mediating voltage and Ca²⁺ dependent channel opening, respectively, have to merge eventually" These two statements seem contradictory.

- The abbreviations "DM1" and "DM2" for "double mutants" should be eliminated and the actual double mutants should be put on Figure axes, for clarity.
- Table S2 is very sloppy
- For experimental measurements, from where was BC5 acquired or how was it synthesized? Also, the full chemical name of BC5 (IUPAC-nomenclature) should be mentioned.
- in some places the reader can see "BK channels" and in other they are introduced simply as "the BK channel". How many "BK channels" are there? Please add the gene name of the BK channel and be consistent throughout the paper with the formulation; perhaps "BK channels" should be preferred since it is the formulation in the title.

Reviewer #1:

This work by Zhang et al explores the mechanisms underlying allosteric modulation of pore opening in BK channels, which remains a key yet unanswered question in the field. Several findings in recent years, including the elucidation of Ca²⁺-bound and Ca²⁺-unbound structures and other functional studies (well recognized in the manuscript), point to a relevant role of the CTD-VSD interface in these mechanisms. Specifically, Zhang et al identify a novel molecule by in silico screening, BC5, that interacts dually with the channel, producing two functional effects: blocking of outward currents, possibly by interaction with the pore, and activation of tail currents, by interacting with the VSD-CTD interface, at a location coincident to great extent with the regulatory Mg²⁺ binding site. Consistent with this fact, the authors demonstrate that BC5 partially competes with Mg²⁺ for the binding site, and mutations abolishing Mg²⁺ activation also partially abolish BC5 activation effects. This is further confirmed by MD and site mutation analyses (but see some comments below). Quantification of gating currents shows that VSD function is not altered by BC5, whereas intrinsic opening is increased, leading the authors to conclude that even though they bind at close locations, the BC5 activation effect involves a mechanism different from that of Mg²⁺, specifically by modulating the Ca²⁺-dependent pathway. Further experiments are performed using different Ca²⁺ concentrations and BK constructs with impaired Ca²⁺ binding, to propose that BC5 may interact with the Ca²⁺-binding-to-pore-opening molecular pathway, which is also supported by specific mutations of residues potentially involved.

The work is solid, experiments are well planned and performed and the manuscript is very well written and structured. The topic is highly timely and provides interesting ideas, setting the background to test intriguing new hypothesis related to BK function and, more generally, allosteric protein function. I believe that it will be of great interest to the extended scientific community. Some concerns, comments and suggestions are listed below.

Line 155 and Fig. 2: Regarding the results with mutant E99R, the authors show that the V-half shift produced by BC5 is reduced (although not to a great extent) by the mutation, consistent with the idea that BC5 and Mg²⁺ bind to at least partially common sites. Does the inhibitory effect of BC5 remain the same in this mutant? If this is the case, this would further support the idea proposed by the authors that the inhibitory and activation effects of BC5 on BK function are unrelated.

Yes, BC5 inhibition effect remains in the presence of Mg²⁺. (Please see sFig. 3a).

The authors mention that the E399 residue is also shared by both Mg²⁺ and BC5 binding sites (line 150). This residue is mutated to R as shown in Fig. 2f, with apparently very different effects that the potentially analogous mutation of D99 to R

shown in Fig. 2c. It is true that EC50 is increased, but the V-half shift seems to be increased as well (this also applies to DM2 containing the same mutation. Data related to mutant D99R are not included in Fig. 2f so comparison is not possible. Can this be better clarified?

We thank the reviewer for asking this question. It shows that we didn't make the distinction between D99 and E399 clear in the manuscript.

D99 is not part of the binding pocket for BC5 from our docking and molecular dynamic simulations. However, it is located close to the binding site and that is why mutation D99R can affect BC5 binding by electrostatic interaction as shown in Fig. 2c. We did not include D99R results in Fig. 2f because Fig. 2f only plots the results for the BC5 binding residues. We have added "*although D99 is not part of the BC5 binding pocket*" in the text (page 8, 1st paragraph) to make these distinctions more clear.

E399, on the other hand, is in the BC5 binding pocket. Consistently, the mutation E399R did increase EC50 for BC5. BC5 binding to the channel is dynamic with many interchangeable modes (Fig 2d). The mutation E399R may prevent BC5 from some of binding modes but not others, which may explain the increase of the maximum GV shift caused by BC5 binding. This effect is discussed in the revised manuscript (page 9, 1st paragraph): "*Since BC5 binding to the channel with multiple modes, a mutation to a single residue in the binding pocket may not be sufficient to abolish BC5 binding. Instead, the mutation may prevent some of the binding modes but reinforce others. This may have happened to some of the mutations, which, while enhancing EC50, also increased the slope of the BC5 dose response and the maximum channel activation (Fig 2e, 2f). To further disrupt BC5 binding, we made double, triple or quadruple mutations that combined these individual mutations. However, except for the two shown in Fig 2e and 2f, most of these combined mutations abolished channel function. These results suggest that this pocket of residues are important for channel function. Consistently, we found that some of the individual mutations altered channel function (sFig 5).*"

Figure 2f: what is the rationale for double mutations DM1 and DM2?

We tried to combine mutations of binding residues to reduce BC5 activation effect more. But none of triple mutations or quadruple mutations had good expression. Only these two double mutations expressed well, and we showed results. The results of the double mutations were not equivalent to a linear additive effect of the two individual mutations. This could be due to dynamic nature of BC5 binding and the complex changes in BC5 binding modes caused by mutations. The rationale and results of double mutations are discussed in the revised manuscript (page 9, 1st paragraph, please see response to the above comment).

Figure 2a: This representation may mislead some unaware reader. Although it is understood that Mg²⁺ and BC5 cannot bind simultaneously to the binding site, perhaps a different representation could be used to make this aspect perfectly clear, if this is the case.

Thanks for the suggestion. We changed the Mg²⁺ symbol to a hatched circle and revise the figure legend to clarify that Mg²⁺ and BC5 do not bind at the same time.

Line 153: "BC5 at the same concentration". Not clear what concentration is this referring to, same as Mg²⁺? What is the criterion to use this concentration for comparison with Mg²⁺ effect?

We now put "100 μ M" after "the same concentration". In this study, we always used a saturating BC5 concentration of 100 μ M unless otherwise noted.

The authors propose that the interaction site is very similar to that of Mg²⁺ regulating the channel, however they function in a radically different way. The difference between underlying mechanisms is merely described in the text but I find that this aspect could be further discussed.

We added a sentence to remind the readers that BC5 and Mg²⁺ binding sites are located closely in the text before the description of their difference in mechanisms (page 9, 2nd paragraph): "*Since the BC5 and Mg²⁺ binding sites are located closely and share E399 (Fig 2a), we examined if BC5 activates the channel via a similar mechanism as Mg²⁺ by modulating VSD activation³⁶.*" We hope that this would emphasize the contrast.

Fig. 2f: No statistical data is provided here. Would it be possible to add eg the confidence intervals?

We re-fitted the curve and provided standard errors in Fig. 2f as suggested. We revised the figure legend accordingly. The curve fitting method is provided in Materials and Methods (page 20).

The data in Fig. 2f were obtained from fitting the data in Fig. 2e with the Hill equation. Due to experimental difficulties, the data at different BC5 concentrations could not be collected on the same oocyte expressing the mutant channel. Instead, each data point was an average of several experiments (n = 3-7) on different oocytes, while each of the oocytes may be measured at one or several BC5 concentrations depending on how well the oocyte could endure the experiments. In the end, all the data for each mutant channel were plotted and fitted in Fig 2e to obtain the EC50 and

Max $\Delta V_{1/2}$ in Fig 2f. Although we could obtain the standard errors, we cannot use any meaningful statistics to estimate the significance of changes.

Regarding the dynamic network and information flow analyses (Fig. 4, line 214 and beyond): This analysis does not include the RCK2 site. Is this assuming that the activation pathway from Ca^{2+} binding to RCK2 is common to, and/or necessarily includes activation of the RCK1 site? How does this relate to the findings of Lorenzo-Ceballos et al (Ref 33 in the manuscript)?

This is an excellent question. We found that BC5 affected the pathway linking the pore to both the RCK1 and RCK2 sites (sFig. 10). In the revised manuscript, we further analyzed the coupling pathways from RCK2 to the pore. The result, shown in a new sFig. 11, demonstrate that the part of the pathway downstream from the BC5 site to the activation gate is common to both Ca^{2+} binding sites, but the part of the pathway upstream from the BC5 site to the two Ca^{2+} binding sites diverge to connect individual Ca^{2+} sites.

A short discussion is now included on Page 15: *“The dynamic pathway analysis shows that the part of the pathway downstream from the BC5 site on the VSD-CTD interface to the activation gate is common to both Ca^{2+} binding sites, but the part of the pathway upstream from the BC5 site to the two Ca^{2+} binding sites diverges to connect individual Ca^{2+} sites (sFig. 11).*

With regarding to the results in relation to the findings of Lorenzo-Ceballos et al, we think that the CTD part of the pathway reported here is specific to Ca^{2+} dependent activation, and should not be related to their results. The findings of Lorenzo-Ceballos et al are about the interaction between Ca^{2+} binding and voltage sensor activation, which is not observed in this study since BC5 did not alter voltage dependent gating.

Text should be revised for typographical errors. Some examples:

Line 25: spaniing

Line 31: mutagenesis

Line 269: patheay

Thanks for pointing these out. Corrected.

Reviewer #2 (Remarks to the Author):

The ms by Zhang et al addresses the mechanism of a novel BK channel activator, BC5, identified from a relatively small subset of compounds (40K) by an in silico screen that

targeted the VSD-CTD interface. BC5 was observed to elicit a shift in the G-V curve toward negative voltages, determined from tail current measurements. At the same time, BC5 was observed to strongly block outward current at positive voltages. The activating and blocking effects may arise from separate mechanisms, because mutations targeting the presumed binding site at the VSD-CTD interface can reduce the activating effect, while the blocking effect remains intact. BC5 effectively activates channels in the absence of Ca, while BC5 appears to compete with Ca activation. The authors conclude that the VSD-CTD interface is critical for transducing the Ca-activation mechanism from the CTD to the pore.

Overall the ms is generally well written and the experimental data are of high quality. The discovery of a new pharmacological probe of BK channels targeted to the VSD-CTD interface will be of interest. There are, however, some concerns that should be addressed in the interest of improving the presentation.

1) It would be worthwhile to present some more details of the *in silico* screen. Prior to screening, was the target binding pocket identified arbitrarily, or was a computational approach used for this as well? Also, why was the screening space limited to charged molecules?

Thanks for the valuable suggestion. We added the details of the *in silico* screening method in the revised manuscript (see Pages 16, 17).

Regarding the first part of the question, both open (Ca²⁺-bound, PDB entry: 6v38) and closed (Ca²⁺-free, PDB entry: 6v3g) conformations of the human BK channel have been solved by using Cryo-EM. A major difference between these two conformations is the relative position between the membrane spanning region (S0-S6) and the cytosolic domain (CTD), as shown in the newly added SFig. 1a. In this study, we aimed to discover chemical compounds stabilizing the BK channel during its open state. Therefore, the interface between the voltage sensor domain (S1-S4) and CTD was selected for the target site and the channel structure in the open state (PDB entry: 6v38) was used in the *in silico* screening study (see Fig. 1a and the new sFig. 1b).

Regarding the second part of the question, we restricted *in silico* screening to the charged molecules for two reasons. First, the VSD-CTD interface contains many charged residues (e.g., K98, D99, D173, and E219 on VSD, and R342, E374, K392, and E399 on CTD). Second, due to remaining limitations of the existing scoring functions (ITScore2 in the current study), we elected to screening only charged molecules to increase the hit rates of the *in silico* screening.

2) It is not entirely clear that the activating effect of BC5 is completely separate from the apparent pore-blocking effect. Isn't it possible the BC5 could be binding in the pore and stabilizing the open state, thus slowing the deactivation kinetics (as shown in Fig S1)? Are single channel open times at negative voltages increased in the presence of

BC5? Although Po is shown in Fig 3, it is not clear whether the open times are longer. If so, this would further support the idea that the drug is activating the channel under conditions where it is not blocked.

We found that inhibition and channel activation are two different processes based on multiple lines of evidence.

First, the dose-response curves for inhibition and activation are different. The IC50 for the inhibition is less than EC50 for the activation.

Second, the inhibition and activation have different voltage and Ca²⁺ dependence.

The inhibition depends on voltage, but not on Ca²⁺. At low and high Ca²⁺, inhibition is the same. On the other hand, the activation depends on Ca²⁺, it can activate the channel only at low concentrations or in absence of Ca²⁺.

Third, mutations that reduced activation did not alter inhibition.

In addition, Mg²⁺ application and mutation results shown in Fig 2 support that BC5 activated the channel by binding at the VSD-CTD interface but not in the pore.

Having these said, we appreciate the point raised by the reviewer on whether the slowing of deactivation was due to BC5 binding in the pore to stabilize the open state. Previous studies showed that Ca²⁺ activates BK channels by reducing deactivation rate, increasing activation rate and shifting GV to more negative voltages (Cui et al., JGP 1997). Since BC5 activates BK channels by perturbing the Ca²⁺ dependent activation pathway, it is consistent that BC5 activated the channel also by reducing deactivation rate and shifting GV to more negative voltages. On the other hand, in Fig. S2 the slowed deactivating kinetics can be readily observed at the negative voltages in the high concentrations of BC5 that block the channel at positive voltages, indicating that current is flowing through channels at the negative voltages but not at the positive voltages. If current is flowing at the negative voltages, there is little if any BC5 in the channel to stabilize the open state at the negative voltages, so the stabilization has to be through a different mechanism, such as BC5 binding to a different binding site not in the pore. Note that BC5 is so large that if it were in the conduction pathway (pore) stabilizing the open state, then it would be expected that the currents would be greatly decreased in amplitude so that the slowed deactivation currents would not be observed, yet they were observed. Therefore, the observation of the deactivation currents at negative potentials indicates that BC5 is not in the pore at the negative potentials, because if it were, then currents would be blocked and not observed. If BC5 is not in the pore at the negative potentials, then the slowed deactivation currents cannot be attributed to BC5 stabilizing the open channel from pore block, so the slowed currents would result from BC5 acting at some other site.

Another argument that BC5 can act to increase P_o when BC5 is not blocking the pore is the observation in Fig. 3b that the BC5 induced increase in P_o appears independent of the test potential. If the increase in P_o were from pore block, then the increase in P_o by BC5 should have increased as the test potential became less negative because the amount of BC5 in the pore should have increased as the test potential became less negative. This was not observed. Thus the effect of BC5 appears independent of pore block because the 5 fold increase in P_o remained constant for the wide range of test potentials. If not pore block, then action at some other site.

The time of single opening events in Fig 3 could not be reliably analyzed due to brief openings and relatively small number of fully open events.

3) I'm not sure whether it is appropriate to call BC5 an activator/agonist, because the net effect of this compound under physiological conditions is to block outward current, and the blocking effect is actually stronger than the activating effect (with a lower IC_{50}). The authors may wish to re-word the title and take out the term "allosteric agonist".

Thanks for this good point. We changed "allosteric agonist" to "allosteric modulator".

Reviewer #3 (Remarks to the Author):

The paper presents the identification of a single compound which alters BK channel physiological parameters in an interesting way. The introduction is unclear and there are important pieces of information missing from the methodological section, affecting the reproducibility of the results and suggesting that the binding pocket could have been wrongly identified. The experimental procedures could have been carried further to inquire on the pharmacological properties. The importance of the finding is not fully explained in the therapeutic setting. Major improvements are needed to raise the manuscript to the standards of the journal, to allow full result reproducibility and to put the discovery in perspective.

We have revised the Introduction and Methods sections as suggested.

Major comments

-The introduction lacks a logical flow. The second paragraph from the introduction starting at line 58 is very confusing to the reader. This paragraph discusses the known mechanisms of activation and should be placed after the BK channel elements (especially the sensors) have been put into structural context. The paragraph must present in a more clear and concise way the two sensors. Take for instance the phrase "In these activation mechanisms Ca^{2+} or voltage alters the conformation of its sensor" what sensor are we talking about? Ca^{2+} and voltage at once or each other individually

are able to “alter” the sensor(s)? Next phrase “although Ca²⁺ and voltage activate BK channels via distinct mechanisms, the two activation mechanisms influence each other that can be described with an allosteric linkage” What are the two activation mechanisms? Please introduce them very clearly after the structural explanation. Then explain the intertwining between them (e.g. Ca-activation is independent of the VSD state).

We revised Introduction as suggested by the reviewer. We described the structural features and the Ca²⁺ and voltage sensors first in the revised second paragraph (page 3, 4).

-The third paragraph of the introduction should be split in two, as it is too large. As mentioned, one paragraph should be devoted to the structural features of the BK channel. Such explanation was attempted in this third paragraph. The second part of the third paragraph, as I see, is more focused on the detailed intricacies of channel activation and should be placed after the general description of how the two sensors/activation mechanisms work. The clarity of these ideas must follow, with the reader in mind.

Revised as suggested (see page 4, 5). We thank the reviewer for this and the above comments. We rearranged these paragraphs and find that it made Introduction much clearer.

-The Mg-based activation is firstly introduced very late in text, in line 93. There are no previous references to the Mg-activation possibilities manifested by BK, nor an explanation of the binding sites of Mg⁺⁺ vs Ca⁺⁺.

Mg²⁺ dependent activation is now more clearly introduced in paragraph 3 of Introduction (page 5): “...because residues from both the CTD and VSD of the neighboring subunits form a Mg²⁺ binding site, which differs from the Ca²⁺ binding sites in CTD. Mg²⁺ binding activates the channel by enhancing voltage dependent activation via an electrostatic interaction with the voltage sensor³⁶.”

-All the structural features presented must be accompanied by placement of flanking or important amino acids (residue numbers, first and last of various sections of the protein) and by a small Figure overviewing the main structural elements of the channel, in both monomeric and multimeric forms. Such a Figure may include both 3D renderings and a 2D membrane-spanning view and could be included in Figure1. Multiple references in the text could be made to this figure.

We agree that structure features should be presented with an overall context that allow the readers to see clearly which region(s) of the channel these features come from.

Following the reviewer's suggestions, we have added the main structural elements of the channel in both 3D rendering and 2D membrane-spanning views in a new sFig. 1a. The important interacting amino acids and their residue numbers are plotted in Fig. 2, panels (a) and (d).

We have also gone through all figures and determined that adequate context is provided for Fig 1. Nonetheless, we have revised Fig 1 caption to further note that " ... *The CTD and VSD are colored pink and cyan, respectively. BC5 is colored red and represented as molecular surface.*"

For Fig 2 & 4: we have modified the caption to note that Fig 2a, d and Fig 4a, b show a similar region as illustrated in Fig 1a.

-throughout the body of the paper there is not an instance mentioning the two regulators of potassium conductance (RCK), parts of the cytosolic domain. Yet, at figure S8, the mutation D367N is said to ablate the Ca-binding site in RCK1.

This is a good catch. RCK is now defined in the second paragraph of Introduction (page 4): "...*CTDs, each contains two structural motifs known as the regulator of potassium conductance (RCK),...*"

-the molecular docking Method is poorly reproducible (to the extent of the impossibility of reproduction). There is no fixed mandatory word limit for the Methods section in Nat Commun (the informal one is 3000 words but this can be exceeded and the section in the paper spans about 2000 words) thus there is room for improvement. Has the Mdock piece of software ever been used in other similar virtual screening situations? Please provide references if so. If not, please state why it is optimal for such purpose. Does Mdock have any settings that can be tweaked by the user? What parameters were set for docking algorithms? Overall, the virtual screening is a little too straightforward and this is not acceptable, particularly given limitations of *in silico* approaches as compared to experimental screens (please compare the process of virtual screening to the one mentioned in various papers e.g. Chemical Physics Impact, Volume 3, December 2021, 100048) and could have been accompanied by a larger experimental screen (of more than 9 compounds).

Thanks for the helpful suggestion. We rewrote the *in silico* screening method to provide the details (see Pages 16, 17).

-What is the "Available Chemical Database" (ACD, Molecular Design Ltd.)? I cannot seem to find it anywhere, nor can I find information about the company. Please provide links and mentions as to how any researcher can reach this Database. Has it been used in other works from the Authors or other researchers? If yes please cite

accordingly. Why was this Database selected (I assume it is not free) and not freely available ones derived from renowned PubChem or ZINC (time-proven, with thousands of citations)? Does it possess any advantage? Does it cover the diversity of the chemical space accordingly?

The Available Chemical Database (ACD) was provided by Molecular Design Ltd., which is currently a part of BIOVIA in Dassault Systèmes, <https://www.3ds.com/> (see Page 17). Please refer to https://en.wikipedia.org/wiki/MDL_Information_Systems for details of ACD. We also screened this chemical library for other ion channels, such as the KCNQ1 channel (*PNAS* **118**: e2024215118, 2021; *Commun. Biol.* **3**: 385, 2020) (Page 16).

Because accurately predicting protein-ligand binding energies with the existing scoring functions remains a challenging task and because we attempted to reduce the cost by purchasing and experimentally testing fewer than 10 compounds, we focused on an enriched subset of compound database in this proof-of-concept study. Namely, our aim was to discover a chemical compound targeting this novel binding site (the VSD-CTD interface) and study the calcium activation mechanism of the BK channel. In future studies, we will screen large, diversified chemical libraries targeting this site to find more potent and efficient compounds than BC5.

-Was the flexibility of the protein residues accounted for in the virtual screening?

The protein structure was treated as rigid bodies in this virtual screening study (see Page 16).

-What program was used to handle the PDB file(s)?

The UCSF Chimera was used to prepare the PDB file (Page 17).

-What is the rationale for selecting PDB version 6V38 (Ca-bound) and not other states (e.g. Ca-free) for docking?

We have addressed this important question in the revised manuscript (Page 16). Specifically, a major difference between the two conformations (open conformation with Ca-bound and closed conformation with Ca-free) is the relative position between the membrane spanning region (S0-S6) and cytosolic domain (CTD), as shown in the newly added sFig. 1a. As we aimed to discover chemical compounds stabilizing the BK channel during its open state, we therefore selected the interface between the voltage sensor domain (S1-S4) and CTD as the target site and selected the structure in the open state (PDB entry: 6v38) in the *in silico* screening study (see Fig. 1a and the new sFig. 1b).

-Where was the searching space located? Page 6 line 123 states is to be “a site near the VSD-CTD interface” but this is very insufficient for reproducibility. What was the center and the radius of the searching space (provided it was a sphere, but this is not mentioned)? What is the rationale for removing the Mg ion from the PDB file? Line 148 at page 7 states that “docking site of BC5 in BK channels is located near the Mg²⁺ binding site”, so does not this mean that it does not overlap with the Mg-binding site? Is it just “near” or is the docking site overlapping with the Mg-binding site? Should the ligand prevent Mg binding?

To address this question, we added sFig. 1b to show the target site, which locates at the VSD-CTD interface. In the PDB entry 6v38, a Mg²⁺ ion is located at the VSD-CTD interface and the residues interacting with Mg²⁺ are included in our screening site. This Mg²⁺ ion was removed before docking (see Page 17).

Since the screened ligands may share some interacting residues with Mg²⁺, they may interfere with each other. As described in this manuscript (Page 8), “we found that BC5 at the same concentration (100 μM) induced less shift in GV relations in the presence of 10 mM Mg²⁺ (Fig 2b, c, sFig 3a). Likewise, a charge reversal mutation of the Mg²⁺ binding residue, D99R, also reduced BC5 activation of the channel (Fig 2c), although D99 is not part of the BC5 binding pocket, supporting that BC5 binds in the vicinity of the Mg²⁺ binding site.”

-What is the rationale for selecting a subset with charged compounds? Is the subset formed solely based on this criterion (charge +1 or +2) or were there other criteria for selecting the subset? How many compounds were there in the initial set?

Because the target site (VSD-CTD interface) contains many charged residues (e.g., K98, D99, D173, and E219 on VSD, and R342, E374, K392, and E399 on CTD), using a chemical library with compounds containing charged groups would increase the hit rate of the *in silico* screening and reduce the experimental cost on compound purchase. Therefore, we considered a subset of compounds in which each molecule has one or two charged groups (a total of ~ 4×10⁴ compounds). We described the rationale in revision (page 17).

-In which sense were the top candidates “manually inspected” (I assume “visually” would be a better choice of words)? Was the virtual screening carried out multiple times to ensure a statistical analysis of variability? Exactly how many compounds were in the top 2% of the ranked candidates? I assume all the compounds were ranked. This leads to 2% of the 40000 compounds, which is about 800 compounds, from which only 9 were “manually” chosen? The selection of 9 compounds out of 800 is perhaps the most important substep and must be accurately described. Nonetheless, it does not seem feasible nor practical. A supplementary table or figure of the process of selecting

the compounds is needed (such as an workflow or pipeline etc.), which may include the predicted binding free energies from Mdock. Based on what criteria were the 9 candidates selected?

We agree that “visually” is a better word than “manually”. We visually inspected the docking modes of ranked compounds (evaluated by the scoring function, ITScore2). Compounds forming strong interactions such as hydrogen bonds or salt bridges with both VSD and CTD were selected. One of these compounds, BC5, forms hydrogen bonds with N172 and D173 from VSD and S337 and E399 from CTD, respectively. Details are described in the subsection entitled “*in silico* screening” in the Methods of the revised manuscript (page 17).

-Are all compounds in the subset commercially available? What can be stated of the other eight compounds, about their names, structures and computational or experimental features? The docking-based screen is the first and foremost step of the methodology, yet it is very poorly described and BC5 seems to be a serendipitous result. If only 1 one of the 9 candidates was positive for your interest, why would the other 800 or so not be eligible for even greater impact?

Some compounds in the subset were not commercially available. In the last step of *in silico* screening, only purchasable compounds were selected for experimental assays. sTable 3 reports the chemical structures, names, MDL, molecular weights and docking scores for the 9 tested compounds.

As discussed previously, because accurately predicting protein-ligand binding energies with the existing scoring functions remains a challenging task and because we attempted to reduce the cost, we visually selected, purchased and assessed only 9 compounds. Our goal was to discover a chemical compound targeting this novel binding site (the VSD-CTD interface) and study the calcium activation mechanism of the BK channel. Indeed, there could be more compounds in the top ranked list that perform better than our selected compounds. In future studies, we will screen large, diversified chemical libraries targeting this site and assess more compounds in our top ranked list to find more potent and efficient compounds than BC5 (see Page 17).

-How exactly were the interaction parameters assigned to the ligand for MD simulations? What tool was used to do this tedious task? What “similar small molecules” from the CHARMM36m force field were selected? The CHARMM36m is a force field for proteins and other tools have to be used, such as CGenFF or SwissParam etc. Parameterizing a small molecule for MD simulations is a very complex process and prone to errors, which is why automated tools combined with (if needed) manual adjustments are needed.

Thanks for the question. As recommended by Charmm36CGenFF (J Comput Chem.

2010 Mar; 31(4): 671–690: ‘Emphasis in the development of the force field was placed on supplying highly optimized chemical building blocks that users can assemble into their molecules of interest.’), we generated the topology for BC5 using existing CHARMM36cgenff and CHARMM36m building blocks. An additional SI figure (sFig. 12) is provided to annotate the choice of building blocks. Standard CHARMM36m parameters are used for all atom types and interactions involved in BC5. A short discussion is also included in the Method section (page 21), “*The interaction parameters of BC5 were then assigned by identifying similar atom, bond, angle, and dihedral-angle types from similar small molecules within the CHARMM36m all-atom force field and CHARMM General Force Field (Charmn36cgenff) (see sFig 12).*”

-Also, was the small molecule manipulated in any way in some software? In what format were its coordinates? Was it built from scratch or it was in pdb format from the start? Did CHARMM-GUI accept it as an input without any modifications?

The initial coordinates of BC5 in complex with the channel were generated from the docking study and in pdb format). Because BC5 is not a standard ligand recognized by CHARMM-GUI, we first generated a fully solvated protein-membrane system without BC5. BC5 was then added to the system using in-house CHARMM scripts. This was noted on Page 21: “*The initial binding pose of BC5 was identified with docking study. In simulations 1-3 and 6-9, a BC5 molecule is placed in binding pocket in each of the four subunits.*”

-CHARMM-GUI already provides simulation parameters (input configuration files). Please provide more clear information on what was left at default and what was changed and why.

This is a good point. We have revised the “Method: Atomistic Simulations” section to clarify that “*The default CHARMM-GUI generated configuration was used except otherwise noted here.*” (see Page 21).

-There is insufficient proof to ascertain that the binding site is actually correctly identified. The ligand “wobbles” around during MD simulations and has zero stability. The docking method is used to identify stable binding poses of ligands; the MD actually shows this is not the case thus defeating the whole purpose of docking. Basically, the MD does not add new insights to the paper as it only proves the ligand “stays” in a presumed binding pocket. It is well known that stability of the ligand during MD simulations is indicative of its effect on the binding site and protein (J Comput Aided Mol Des 2017 Feb;31(2):201-211). The ligand binding is described simply and thus poorly in plain terms of probability of residue contacts, with poor mentions whatsoever about actual molecular interactions (e.g. H-bonds, pi-stacking etc.) although judging from the presumably high RMSD during simulations all the bonds seem transient. Also,

the effects of the ligand are manifested on a channel with natural allosteric modulation, which makes it even harder to accurately pinpoint a binding site for a ligand. In my opinion, this is a major limitation that should be explicitly mentioned in the Discussion.

BC5 is a weak agonist of BK channels with an IC₅₀ of 2.5 μM (Fig 1e). Many ligands of similar affinities have been shown to interact dynamically with proteins. The most prominent examples are probably from the studies of intrinsically disordered proteins (see Chen et al, *Biomolecules* 10, 743 (2020) for a summary of recent examples from experiment as well as computation). Several blockers and activators of BK channels have also been shown to bind dynamically (e.g., Jia Nature Commun 2019; Schewe et al Science 2019). We believe that an important insight from MD is precisely about the dynamic nature of BC5 interaction with BK channels.

The molecular details of the dynamic BC5-BK interactions are summarized on Page 8, *“As summarized in sTable 2, the hydrophobic naphthalenyl group of BC5 was buried in contact with the intracellular side of S0, S1 S3 and S4 as well as the residue F395 from αB of CTD. The charged arginine part of BC5 interacted with the negatively charged residue E399 (Fig 2d). The binding poses of BC5 were more narrowly confined in the open conformation, suggesting that BC5 binding is stronger and may favor the open state of BK channels.”* In addition, we have expanded sTable 2 to include notes on the nature of BC5 interaction with each contacting sidechains.

Following the suggestion of the reviewer, we added to discussion on Page 15, *“It should be noted that current study suggests the BC5 adopts multiple binding modes in the CTD-VSD interface and most contacts are transient. Thus, further chemical modification of BC5 could yield new compound with more stable interactions with BK channel, which may possess stronger allosteric activation activity.”*

-In line with the previous comment, page 12 line 262 states that “BC5 binds at the CTD-VSD interface”; this may only be certainly demonstrated using a direct physical method such as crystallography, NMR spectroscopy etc. Please use words such as “suggest”, “putative” or “indicate”. Throughout the paper there should be more skepticism about this binding site and the results from MD should be treated as “indicative”; furthermore, it cannot be excluded that the ligand may bind at multiple sites at once!

Changed as suggested. The revised manuscript now states *“The results suggest that BC5 binds at the CTD-VSD interface (Fig 2).”*

-There is no mention of how the eight mutations included in the text and shown in Figures 2e and 2f alter the WT physiological parameters in the absence of the ligand. Are they completely neutral in this regard? This may actually emphasize that the binding site was more correctly identified. Also related, mutation N182A alters agonist's

effects similar to the other five tested single residue replacements, yet this residue does not contact the ligand directly in docking or MD. Please provide explanations for this effect.

This is a good point. We added a supplementary figure (sFig 5) to address this question and discussed the results in the 4th paragraph of Results.

-Please clearly emphasize the importance of the identified compound as compared to other agonists (Molecular Pharmacology December 2, 2020, MOLPHARM-AR-2020-000106; Molecular Pharmacology April 30, 2022, MOLPHARM-AR-2021-000478; 3552–3557; PNAS February 28, 2012 vol. 109 no. 9) in a paragraph in Discussion.

This study focuses on using BC5 as a tool to understand molecular mechanisms of BK channel activation, which is also a general question in protein function. While BC5 is a new BK channel modulator, we feel that a comparison of BC5 to all other BK channel agonists deviates from the scope of the paper. We discussed the potential use of BC5 in the end of Discussion.

-The discussion should be more focused on the novel discoveries made by the Authors, delineating the elements which have not been known previously from other papers.

We aimed to focus our discussion on the molecular mechanism of BK channel activation, i.e., the allosteric pathway for Ca²⁺ binding to trigger pore opening, which is summarized in the first paragraph in Discussion.

-The forte of the paper is not sufficiently emphasized, that of discovering a ligand that has novel properties compared to other ligands. This is what needs to be put into therapeutic context at the end of the paper. There is no ADMET data/prediction about this compound, nor any mentions about its use in other contexts, but could it be useful as a lead compound? Its extensions as a lead compound for future studies could be mentioned, similarly to the recent experimental screening of Srinivasan et al. (Molecular Pharmacology April 30, 2022, MOLPHARM-AR-2021-000478).

Please see our response to the above two comments. This study is to understand molecular mechanisms of BK channel activation, which is also a general question in protein function. While BC5 is a novel ligand of BK channels, it was discovered for the purpose of understanding BK channel gating mechanisms. While we agree with the reviewer that BC5 is an interesting ligand, we would like to focus on the major goal of our study. For this reason we tried not to emphasize BC5 per se.

Minor comments

-there are two words spelled wrong in the Abstract: spaniing and mutagenesis

Corrected.

-the abbreviation CTD stands for "cytosolic tail domain" (not just "cytosolic domain"), please correct where needed

Done.

-page 4 line 73 "fundamental" instead of "foundamental"

Corrected.

-page 3 line 45 "On the other hand" implies that a previous phrase starts with "On one hand" please correct. Same on line 68. Same on line 82.

We checked all places where "on the other hand" was used, and made corrections if the previous phrase could not start with "on one hand".

-page 4 line 61 "alter" instead of "alters"

Changed.

-page 5 line 90 "tugging" should be placed between apostrophes (quotation marks or similar)

Done.

-line 316 the charges cannot be neutral, are either positive +1 or negative -1

We apologize for the confusion. To clarify this issue, we have changed the original sentence "... *each compound has a formal charge of either 1 or 2*" to "... *each compound contains either one charged group (carrying a charge of +1, -1, +2 or -2) or two charged groups (with one group carrying +1 charge and one group carrying -1 charge)*" (page 17).

-line 269 "pathway"

Corrected.

-line 273 "extent"

Corrected.

-Figure 2d there is a thick line ligand drawn in the Figure but it is not mentioned in the legend (only the thin lines should be depicted)

The thick lined ligand was included to show one example pose. We have revised Fig 2 caption to note, "*As a reference, representative BC5 binding poses (closed: at 180 ns from sim 7; open: at 102 ns from sim 1) are also shown in thick bonds.*"

-Line 252 states "It is not clear whether the allosteric path eventually merges with that for voltage dependent activation" while line 68 states "the paths for mediating voltage and Ca²⁺ dependent channel opening, respectively, have to merge eventually" These two statements seem contradictory.

"whether" is changed to "where".

-The abbreviations "DM1" and "DM2" for "double mutants" should be eliminated and the actual double mutants should be put on Figure axes, for clarity.

We appreciate the comment. We tried to follow the suggestion but due to the limited space and font sizes we could not make the suggested change.

-Table S2 is very sloppy

We have cleaned up the formatting irregularities of sTable 2 and added a new column to annotate the nature of interaction between BC5 and each contacting residues.

-For experimental measurements, from where was BC5 acquired or how was it synthesized? Also, the full chemical name of BC5 (IUPAC-nomenclature) should be mentioned.

Listed in Materials and Methods under Electrophysiology.

-in some places the reader can see "BK channels" and in other they are introduced simply as "the BK channel". How many "BK channels" are there? Please add the gene

name of the BK channel and be consistent throughout the paper with the formulation; perhaps “BK channels” should be preferred since is the formulation in the title.

There is only one BK channel gene, *sl α 1*, and the BK channel is formed by Slo1 protein. The first sentence in Introduction now describes BK channels as formed by 4 Slo1 subunits.

We thank the reviewer for thoroughly reading the manuscript and pointing out these mistakes.

Reviewers' Comments:

Reviewer #1:

Remarks to the Author:

All my concerns have been addressed. I have no further comments.

Reviewer #2:

Remarks to the Author:

I am satisfied by the authors' revisions and I have no additional comments.

Reviewer #3:

Remarks to the Author:

The authors have addressed my comments.